# CAFE: Causally-Guided Automated Feature Engineering with Multi-Agent Reinforcement Learning

## Abstract

Automated feature engineering (AFE) enables AI systems to autonomously construct high-utility representations from raw tabular data. However, existing AFE methods rely on statistical heuristics, yielding brittle features that fail under distribution shift. We introduce **CAFE**, a framework that reformulates AFE as a causally-guided sequential decision process, bridging causal discovery with reinforcement learning-driven feature construction. *Phase I* learns a sparse directed acyclic graph over features and the target to obtain *soft causal priors*, grouping features as direct, indirect, or other based on their causal influence with respect to the target. *Phase II* uses a cascading multi-agent deep Q-learning architecture to select causal groups and transformation operators, with hierarchical reward shaping and causal group-level exploration strategies that favor causally plausible transformations while controlling feature complexity. Across 15 public benchmarks (classification with macro-F1; regression with inverse relative absolute error), CAFE achieves up to 7% improvement over strong AFE baselines, reduces episodes-to-convergence, and delivers competitive time-to-target. Under controlled covariate shifts, CAFE reduces performance drop by $\sim 4\times$ relative to a non-causal multi-agent baseline, and produces more compact feature sets with more stable post-hoc attributions. These findings underscore that causal structure, used as a soft inductive prior rather than a rigid constraint, can substantially improve the robustness and efficiency of automated feature engineering.

## 1 Introduction

In high-stakes domains such as chemical process optimization, industrial manufacturing, drug formulation, and energy systems, AI systems must operate under evolving conditions where input distributions shift. Automated feature engineering (AFE) promises to extract useful representations from raw tabular data with minimal manual effort. Yet, most AFE methods rely on statistical correlations, yielding brittle features that fail once distributions change, limiting use in safety-critical settings. For instance, in chemical formulation, a 2°C temperature variation can alter product yield by 15–30%, but correlation-based AFE often attributes outcomes to spurious variables rather than causal drivers. As process conditions evolve, such features lose predictive power despite stable causal mechanisms. This highlights a fundamental limitation: conventional feature engineering overlooks causality, producing models vulnerable to interventions or operational shifts. We study **causally-guided AFE**: automatically transforming high-dimensional process data into features that preserve causal relations with target outcomes. This is key for generalization, transparency, and computational tractability.

Yet, causally-guided AFE introduces several core technical challenges: 1) **Causal discovery under noise and confounding:** Real-world observational data often includes latent confounders and noise, making reliable causal structure learning difficult without strong assumptions. 2) **Combinatorial search explosion:** The space of candidate feature transformations scales exponentially with dimensionality, rendering greedy exploration computationally challenging. 3) **Structured exploration under uncertainty:** Navigating the exponential transformation space requires principled search strategies that favor causal plausibility over spurious correlations. 4) **Robustness:** Features should preserve utility under covariate shifts and mechanism-preserving changes.

The existing methods only partially address these challenges. Most traditional AFE methods rely on greedy exploration and utility-based pruning strategies and can be brittle under shifts Kanter & Veeramachaneni (2015); Horn et al. (2019). Reinforcement learning (RL) methods Chen et al. (2019a); Wang et al. (2022b); Liu et al. (2021); Malarkkan et al. (2025) improve exploration but suffer from sparse rewards and ignore causal structure. Recent LLM-based methods Gong et al. (2025); Zhang et al. (2024); Abhyankar et al. (2025) treat feature generation as sequence modeling but lack principled causal guidance. While causal feature selection methods Tsamardinos et al. (2003); Yu et al. (2020); Wang et al. (2023); Yao & Ge (2023) identify causally relevant features, they do not synthesize transformed features. This gap motivates a framework that integrates causality with principled feature generation.

**Our Insights: A Causally-Guided RL Perspective for Feature Engineering.** AFE must move beyond statistical correlations to leverage causal mechanisms that remain stable across operational changes. To meet this end, we integrate insights from causal discovery and reinforcement learning: First, causal graphs summarize structural dependencies, providing invariant principled guidance for which variables to transform and how to combine them. Second, RL methods enable adaptive navigation of large, discrete transformation spaces. We operationalize these foundations through three key insights: 1) *Soft Causal Inductive Bias*: Instead of enforcing rigid causal constraints, the causal structure provides soft inductive priors to guide feature transformation decisions, maintaining flexibility under causal uncertainty; 2) *Causal-aware RL Exploration*: Clustering features by causal roles (direct, indirect, or unrelated to the target), enabling reduced search complexity while preserving causal interpretability; 3) *Causally Shaped Rewards*: Reward functions that balance predictive utility with causal consistency and transformation diversity, ensuring intervention stability.

**Summary of Proposed Solution:** Building on these insights, we introduce **CAFE**, a two-phase Causally-guided Automated Feature Engineering framework. In **Phase I**, we apply sparsity-regularized causal discovery to construct a causal graph over the feature space, categorizing features as direct causes, indirect causes (via multi-hop paths), or non-causal with respect to the target, providing soft inductive guidance for feature transformation policy while maintaining flexibility under causal uncertainty. In **Phase II**, we deploy a cascading multi-agent deep Q-learning architecture where three specialized agents make sequential decisions: selecting causal feature clusters, choosing transformation operators, and constructing causally-informed feature interactions. The agents use hierarchical reward shaping and an adaptive exploration strategy to favor causally plausible transformations while controlling complexity. Together, these phases unify causal reasoning with reinforcement learning to improve robustness and compactness in high-stakes applications.

**Our Contributions:** 1) We formulate AFE as a causally-guided sequential decision process, moving beyond correlation-based heuristics to leverage stable causal mechanisms; 2) We introduce three core principles—soft causal inductive bias, causal structure-aware exploration, and causally shaped reward function, integrating causal discovery with adaptive feature construction through novel multi-agent coordination; 3) We develop **CAFE**, a two-phase AFE framework combining causal graph discovery with cascading multi-agent reinforcement learning that strategically constructs causally-informed feature transformations. 4) We provide extensive empirical validation of CAFE across 15 benchmark datasets, demonstrating consistent improvements up to $7\%$ in predictive performance, reduces episodes-to-convergence with competitive time-to-target, and exhibits $\sim 4\times$ smaller degradation under controlled covariate shifts relative to a non-causal multi-agent baseline.

## 2 PROBLEM STATEMENT

Let $\mathcal{D} = \{(\mathbf{x}_i, y_i)\}_{i=1}^N$ be a dataset with $N$ instances, where each $\mathbf{x}_i \in \mathbb{R}^K$ is a $K$-dimensional feature vector and $y_i \in \mathcal{Y}$ is the corresponding target. Let $\mathcal{F}$ denote the original feature set, and $\mathcal{O}$ a predefined set of transformation operators (e.g., arithmetic, interactions; unary or binary with standard domain guards). Applying transformations from $\mathcal{O}$ to subsets of $\mathcal{F}$ induces a set of candidate features $\mathcal{F}^g$. The goal of AFE is to construct an optimal feature set $\mathcal{F}^* \subseteq \mathcal{F} \cup \mathcal{F}^g$ that maximizes a task-specific performance metric $V_A$ for a given ML task $A$ (e.g., classification, regression). We formulate the objective as:

$$\mathcal{F}^* = \arg \max_{\mathcal{S} \subseteq \mathcal{F} \cup \mathcal{F}^g} V_A(\mathcal{S}, y) \tag{1}$$

where $\mathcal{S}$ denotes a candidate feature subset and $V_A$ quantifies model performance on task $A$. The central challenge lies in efficiently navigating the exponentially large transformation space to identify $\mathcal{F}^*$ that achieves: (i) high predictive utility, (ii) robustness under distribution shift, and (iii) computational tractability in exploring the exponential feature space, guided by causality.

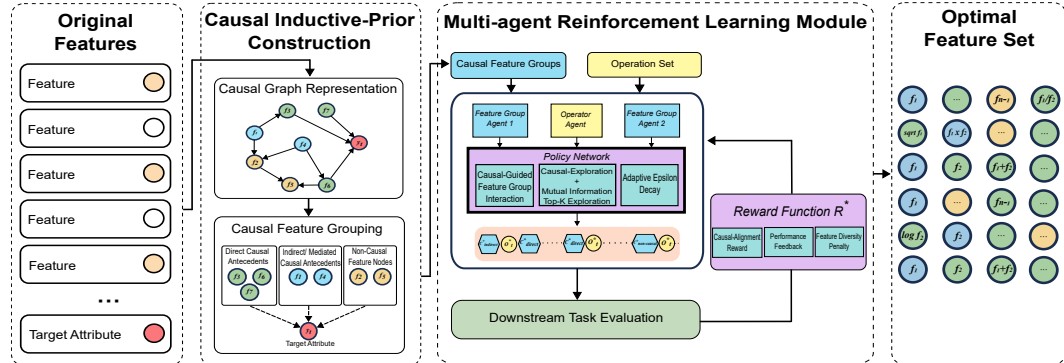

Figure 1: **CAFE Framework Overview.** Phase I learns a causal graph and derives soft causal priors that group features by their relation to the target: direct causes, indirect causes, and non-causal features. Phase II employs cascaded multi-agent reinforcement learning with causally shaped rewards and adaptive exploration strategies, balancing causal coherence with statistical discovery.

## 3 METHODOLOGY

Automated feature engineering under distribution shift must move beyond correlation-based heuristics to exploit stable causal mechanisms. CAFE addresses this via a two-phase framework that couples causal discovery with multi-agent reinforcement learning (Fig. 1). *Phase I* learns a causal graph to group features by relation to the target, providing soft inductive guidance rather than rigid constraints. *Phase II* uses a factorized multi-agent policy to explore the exponential transformation space, steering by causal priors while remaining flexible under causal uncertainty. Causal guidance enters through probabilistic selection and reward shaping; only safety/validity checks (e.g., guarded division) are enforced as hard constraints. Together, these phases yield compact, transparent feature sets that retain predictive performance under mechanism-preserving covariate shifts. By using the learned causal structure as a soft inductive prior, we transform the AFE problem from a broad, undirected search into a more strategic and efficient exploration, justifying the initial computational overhead of causal discovery with faster convergence to high-quality feature transformations.

### 3.1 THEORETICAL FOUNDATION: WHY CAUSAL STRUCTURE GUIDES ROBUST FEATURE ENGINEERING

**Core principle.** In a structural causal model (SCM) with assignments $X_j := f_j(\mathrm{PA}_j, \epsilon_j)$, the conditional mechanisms $P(X_j \mid \mathrm{PA}_j)$ remain invariant across environments even when marginals $P(X_j)$ shift Peters et al. (2016). This invariance suggests that features constructed from variables that are causally relevant to $Y$ are more likely to preserve their predictive relation to $Y$ under mechanism-preserving shifts than features relying on spurious correlations.

**Formal setting.** Consider a structural causal model (SCM) with variables $\mathbf{X} = (X_1, \ldots, X_d)$ and target $Y$, where each variable follows $X_j := f_j(\mathrm{PA}_j, \epsilon_j)$ with parents $\mathrm{PA}_j$ and noise $\epsilon_j$. A mechanism-preserving shift transforms the distribution from $P(\mathbf{X})$ to $P'(\mathbf{X})$ while preserving conditional distributions $P(X_j \mid \mathrm{PA}_j) = P'(X_j \mid \mathrm{PA}_j)$.

**Proposition 1** (Informal invariance for causally sufficient summaries). *Let $S \subseteq \mathcal{F}$ contain a subset $S^\star$ that suffices for predicting $Y$ across environments (e.g., $S^\star \supseteq \mathrm{PA}_Y$ and the shift is mechanism-preserving). If a transformation $\phi$ preserves the information in $S^\star$ (e.g., is injective on $S^\star$ or is a sufficient statistic for $S^\star$), then*

$$P(Y \mid \phi(S)) = P'(Y \mid \phi(S)).$$

*If $S$ excludes any such invariant set, no general invariance guarantee is available.*

*Discussion.* Proposition 1 restates a standard invariance intuition Peters et al. (2016): conditioning (directly or via sufficient transforms) on observed causal parents of $Y$ stabilizes the conditional across mechanism-preserving environments. We give precise assumptions and proofs for idealized settings in Appendix A, and enumerate violations (latent confounding, selection bias, mechanism changes) that break invariance.

**Empirical regularity.** When causal discovery yields a reasonable approximation to the true relations, features built from variables with stronger (estimated) causal connection to $Y$ empirically exhibit greater stability under mechanism-preserving shifts than features built from purely correlational signals. We quantify this trend via controlled robustness experiments (Sec. 4.2.3) and sensitivity analyses under graph misspecification (Appendix B).

This perspective motivates organizing features by approximate causal relationships: direct causes ($X_j \rightarrow Y$), indirect causes ($X_j \rightsquigarrow Y$ via paths of length $\geq 2$), and other variables. This causal hierarchy acts as a soft inductive bias that guides but does not constrain feature construction decisions.

## 3.2 PHASE I: CAUSAL GRAPH DISCOVERY AS A SOFT INDUCTIVE PRIOR FOR FEATURE GROUPING

In Phase I, we construct a causal graph to organize the feature space by underlying causal relationships rather than statistical correlations. The central insight is that features with direct or indirect causal influence on the target provide more reliable guidance for feature construction, offering stable inductive signals under mechanism-preserving shifts, whereas correlations may be spurious.

**Causal Discovery Backend.** Given an observational dataset $\mathcal{D} = \{(\mathbf{x}_i, y_i)\}_{i=1}^N$, we construct a Directed Acyclic Graph (DAG) $\mathcal{G} = (\mathcal{V}, \mathcal{E})$ over the variable set $\mathcal{V} = \mathcal{F} \cup \{y\}$, where $\mathcal{F}$ represents the original features. Each directed edge $(u, v) \in \mathcal{E}$ represents a causal relationship from variable $u$ to variable $v$.

We employ NOTEARS with Lasso regularization Zheng et al. (2018) as our primary causal discovery method, though the framework accommodates alternative algorithms (Appendix B). Given dataset $\mathcal{D} = \{(\mathbf{x}_i, y_i)\}_{i=1}^N$ with features $\mathcal{F}$ and target $y$, we learn a DAG $\mathcal{G} = (\mathcal{V}, \mathcal{E})$ over $\mathcal{V} = \mathcal{F} \cup \{y\}$ by optimizing:

$$\min_{W \in \mathbb{R}^{d \times d}} \frac{1}{2N} \|X - XW\|_F^2 + \lambda \|W\|_1 \quad \text{s.t.} \quad h(W) = \text{tr}(e^{W \circ W}) - d = 0 \tag{2}$$

where $W$ is the weighted adjacency matrix, $\lambda > 0$ controls sparsity, $\circ$ denotes Hadamard product, and $h(W) = 0$ enforces acyclicity via a differentiable constraint. This formulation enables gradient-based optimization while maintaining theoretical guarantees about DAG structure. The optimized adjacency matrix $W$ defines the edge weights of the causal graph $\mathcal{G} = (\mathcal{V}, \mathcal{E})$, where larger weights indicate stronger causal influence. This produces a sparse DAG that captures the most significant causal pathways while filtering out weak or spurious connections.

**Assumptions and Scope.** Our main analysis assumes: (1) causal sufficiency (no unobserved confounders), (2) linear additive noise model with sparse structure, and (3) mechanism-preserving shifts. We evaluate robustness to assumptions violation and compare alternative discovery methods like PC Kalisch & Bühlman (2007), GES Chickering (2002), LiNGAM Shimizu (2014) in Appendix B.

**Causal Role Assignment.** From learned DAG $\mathcal{G}$, each feature $f \in \mathcal{F}$ receives a causal role:

$$\mathcal{M}(f) = \begin{cases} \text{direct} & \text{if } f \rightarrow y \\ \text{indirect} & \text{if } f \rightsquigarrow y \text{ (path length } \geq 2) \\ \text{other} & \text{otherwise} \end{cases} \tag{3}$$

This induces causal-semantic groups $\{\mathcal{C}_{\text{direct}}, \mathcal{C}_{\text{indirect}}, \mathcal{C}_{\text{other}}\}$. To balance signal preservation with computational tractability, we apply within-group screening:

$$\mathcal{C}_g^* = \begin{cases} \mathcal{C}_g & \text{if } |\mathcal{C}_g| \leq k_g \\ \text{top-}k_g(\mathcal{C}_g, \text{MI}(\cdot, y)) & \text{otherwise} \end{cases} \tag{4}$$

where top-$k_g$ selects features with highest mutual information with target $y$. Robust fallbacks (Pearson correlation, $\chi^2$ test, stratified sampling) handle cases where MI estimation is unreliable.

These causal roles serve as soft inductive priors and guide feature transformation decisions without rigid constraints. This design maintains flexibility when causal discovery is imperfect while providing principled structure that statistical methods lack.

## 3.3 PHASE II: MULTI-AGENT REINFORCEMENT LEARNING WITH CAUSAL GUIDANCE

Phase I establishes causal-semantic feature groups through theoretically grounded structure discovery. Phase II addresses the exponentially large transformation space ($O(|\mathcal{O}| \cdot |\mathcal{F}|^2)$) via multi-agent reinforcement learning with a Decision Factorization Rationale, where agents select entire causal feature groups rather than individual features.

**Decision Factorization Rationale.** The AFE action space is combinatorially large: $O(|\mathcal{O}| \times |\mathcal{F}|^2)$ for operator set $\mathcal{O}$ and feature set $\mathcal{F}$. Direct optimization suffers from sparse rewards and high variance. We factorize decisions into three stages: (1) select primary feature group, (2) choose transformation operator, (3) optionally select secondary group for binary operations. This factorization substantially reduces the effective action space each agent explores; from joint choices over (group, operator, partner) to three smaller, structured decisions, while preserving semantic coherence through causal organization. Empirically, this reduces value-estimate variance and improves sample efficiency (learning-curve dispersion in Appendix A). We also include a stylized analysis clarifying conditions where factorization reduces estimation variance; we treat it as a modeling advantage rather than a general guarantee.

**Multi-Agent Architecture.** Three specialized DQN agents operate in cascade: *Primary Group Agent* ($\pi_1$): Selects from $\{\mathcal{C}^*_{\text{direct}}, \mathcal{C}^*_{\text{indirect}}, \mathcal{C}^*_{\text{other}}\}$. *Operator Agent* ($\pi_o$): Chooses transformation from operator library $\mathcal{O}$. *Secondary Group Agent* ($\pi_2$): Selects partner group for binary operations

**State Representation Design.** Agents receive context through statistical feature summaries and group-specific information:

$$s_t^{(1)} = \text{NestedStats}(\mathcal{F}_{t-1}) \oplus \text{GroupStats}(\mathcal{C}^*_{\text{direct}}, \mathcal{C}^*_{\text{indirect}}, \mathcal{C}^*_{\text{other}}) \tag{5}$$

$$s_t^{(o)} = s_t^{(1)} \oplus \text{NestedStats}(\mathcal{C}_t^{(1)}) \tag{6}$$

$$s_t^{(2)} = s_t^{(o)} \oplus \text{OneHot}(o_t) \tag{7}$$

where $\text{NestedStats}(X)$ computes a hierarchical statistical summary: first computing standard descriptive statistics (mean, std, min, max, quartiles) across features, then applying the same descriptive analysis to this summary vector, yielding a compact yet informative representation that captures both individual feature properties and their collective distributional characteristics. This approach provides consistent, interpretable state encodings without requiring learned representations, while maintaining sufficient information density for effective RL decision-making.

**Causally-Guided Exploration.** We design three principled feature interaction strategies that integrate causal theory, statistical relevance, and diversity discovery with adaptive scheduling:

*Causal-Hierarchical Feature Interaction*: Prioritizes transformations respecting causal structure. For unary operations, favors $\mathcal{C}_{\text{direct}} \cup \mathcal{C}_{\text{other}}$ over indirect causes (which mediate relationships). For binary operations, emphasizes $\mathcal{C}_{\text{direct}} \times \mathcal{C}_{\text{indirect}}$ and $\mathcal{C}_{\text{direct}} \times \mathcal{C}_{\text{direct}}$ combinations.
*Top-$k$ Mutual Information Selection*: Hedges against causal discovery errors by incorporating mutual information rankings, selecting features with strongest empirical associations with target.
*Diversity Sampling*: Prevents premature convergence through controlled randomization, ensuring adequate exploration of the transformation space.

Strategy weights adapt based on validation performance using exponential moving averages with decay $\alpha$ (Appendix).

**Reward Function Design.** We align optimization with predictive utility and causal principles:

$$R_t = R_{\text{perf},t} \cdot (1 + \alpha \cdot \Psi_{\text{causal},t}) + \lambda_{\text{div}} \mathcal{H}(\pi_t) - \lambda_{\text{comp}} C(\mathcal{F}_t^g), \tag{8}$$

where $R_{\text{perf},t} = \Delta\text{ValidationScore}_t$, $\alpha \in (0, 1)$ controls causal bonus intensity, and

$$\Psi_{\text{causal},t} = \frac{1}{|\mathcal{F}_t^g|} \sum_{f \in \mathcal{F}_t^g} w_{\mathcal{M}(f)} \cdot \text{Usage}(f), \quad \text{with} \quad w_{\text{direct}} > w_{\text{indirect}} > w_{\text{other}} \geq 0.$$

The modulated structure $(1 + \alpha \cdot \Psi_{\text{causal},t})$ ensures causal bonuses amplify positive improvements while proportionally reducing penalties for causally-relevant features, preventing reward hacking while maintaining causal preferences. The algorithm is given in Appendix A 1.

**Robust Feature Generation.** Transformation operators include domain validation and numerical safeguards addressing common failure modes:

*Logarithmic operations*: Apply to $\log(|x| + \epsilon)$ where $\epsilon = 10^{-8}$ for numerical stability.

*Division operations*: Use protected division $x \cdot \frac{\text{sign}(y)}{\max(|y|, \epsilon)}$ with $\epsilon = 10^{-8}$.

*Square root*: Apply $\sqrt{|x|}$ for negative inputs.

*Range clipping*: Bound generated features to $[-10^6, 10^6]$ preventing overflow.

For binary operations, we limit candidate pairs to $\min(50, |\mathcal{C}_1| \times |\mathcal{C}_2|)$ through relevance-based sampling, maintaining computational tractability while preserving quality.

**Learning and Optimization.** Agents train via temporal-difference learning with experience replay and target networks:

$$\mathcal{L}(\theta) = \mathbb{E}_{(s,a,r,s') \sim \mathcal{B}} \left[ \left( Q_\theta(s,a) - \left( r + \gamma \max_{a'} Q_{\theta^-}(s',a') \right) \right)^2 \right] \quad (9)$$

where $\mathcal{B}$ is the replay buffer, $\theta^-$ denotes target network parameters, and $\gamma$ is the discount factor.

**Scalability and Termination.** We implement intelligent pruning and adaptive termination to ensure computational efficiency while maintaining feature quality. Two-stage filtering first applies variance thresholding to remove near-constant features, then uses mutual information ranking to retain top-k features while ensuring minimum representation per causal group. Multiple termination criteria trigger when: (1) validation improvement falls below $\delta$ for N consecutive episodes; (2) feature count exceeds computational limit; or (3) maximum episodes are reached. This design choice makes the group-choice action constant-size ($|G| = 3$), improving exploration stability. With a fixed operator library and capped within-group sampling ($k_g$) and pairing budget $B = \min(50, |C_1| \cdot |C_2|)$, the per-step decision cost is $O(1)$ for group selection and $O(|\mathcal{O}| + B)$ for operator and pairing, independent of raw dimensionality $d$. Overall runtime remains dominated by candidate evaluation, pruning, and downstream model training. Upon termination, we return the feature set $\mathcal{F}^*$, achieving the highest validation performance across all episodes. Complete state representations, network architectures, and hyperparameters are detailed in Appendix C-D for reproducibility.

## 4 EXPERIMENTAL STUDY

We designed our experiments to meticulously address the following research questions: **RQ1: Predictive Utility.** Does CAFE consistently outperform recent AFE baselines in terms of predictive accuracy? **RQ2: Ablation Study.** How do causal priors, exploration strategies, and reward components individually contribute to the performance and efficiency of CAFE? **RQ3: Distributional Robustness.** Can CAFE generate features that are resilient under covariate shifts and maintain predictive accuracy on out-of-distribution data? **RQ4: Interpretability and Compactness.** Are the synthesized features semantically meaningful and causally aligned with the underlying data-generating process?

### 4.1 EXPERIMENTAL PROTOCOL

**Data Description and Evaluation Framework.** We evaluate our framework on 15 diverse public datasets collected from Kaggle, LibSVM, UCIrvine, and OpenML repositories, across two core tasks: (1) classification and (2) regression. The data statistics are given in Table 1. All experiments employ 5-fold stratified cross-validation with fixed random seeds for reproducibility. We report macro-averaged F1-scores for classification and inverse relative absolute error (1-RAE) for regression, both scale-invariant metrics suitable for cross-dataset comparison.

**Implementation Details.** We use XGBoost with default hyperparameters as the downstream model, following established automated feature engineering evaluation protocols. This choice enables fair comparison across all methods while avoiding confounding effects from model-specific tuning. An extended evaluation across multiple model families (Random Forest, Neural Networks, Linear Models) is provided in the Appendix.

**Baseline Methods.** We compare against 10 representative methods spanning three categories:

*Statistical Baselines*: **Original (ORG):** Raw features without transformation. **Random (RDG):** Random transformation application for lower-bound comparison. **Exhaustive (ERG):** Systematic transformation followed by statistical selection.

Table 1: Overall performance comparison. 'C' for classification and 'R' for regression. **Bold** indicates the best result.

| Dataset | Source | C/R | Samples | Features | RDG | ERG | LDA | AFT | NFS | TTG | GRFG | ELLM-FT | **CAFE** |
|---|---|---|---|---|---|---|---|---|---|---|---|---|---|
| Amazon Employee | Kaggle | C | 32769 | 9 | 0.744 | 0.740 | 0.920 | 0.943 | 0.935 | 0.806 | 0.946 | 0.946 | **0.947** |
| SVMGuide3 | LibSVM | C | 1243 | 21 | 0.703 | 0.747 | 0.683 | 0.829 | 0.831 | 0.766 | 0.831 | 0.836 | **0.846** |
| German Credit | UCIrvine | C | 1001 | 24 | 0.695 | 0.661 | 0.627 | 0.751 | 0.765 | 0.731 | 0.772 | 0.775 | **0.793** |
| Messidor_features | UCIrvine | C | 1150 | 19 | 0.673 | 0.635 | 0.580 | 0.678 | 0.746 | 0.726 | 0.757 | 0.757 | **0.774** |
| Ionosphere | UCIrvine | C | 351 | 34 | 0.919 | 0.926 | 0.730 | 0.827 | 0.949 | 0.938 | 0.960 | 0.963 | **0.976** |
| Wine Quality Red | UCIrvine | C | 999 | 12 | 0.599 | 0.611 | 0.600 | 0.658 | 0.666 | 0.647 | 0.686 | 0.685 | **0.707** |
| Wine Quality White | UCIrvine | C | 4900 | 12 | 0.552 | 0.587 | 0.571 | 0.673 | 0.679 | 0.638 | 0.685 | 0.689 | **0.752** |
| Housing Boston | UCIrvine | R | 506 | 13 | 0.605 | 0.617 | 0.374 | 0.641 | 0.665 | 0.658 | **0.684** | 0.671 | 0.681 |
| Airfoil | UCIrvine | R | 1503 | 5 | 0.737 | 0.732 | 0.463 | 0.774 | 0.771 | 0.783 | 0.797 | 0.786 | **0.799** |
| Openml_586 | OpenML | R | 1000 | 25 | 0.595 | 0.546 | 0.472 | 0.687 | 0.748 | 0.704 | 0.783 | 0.801 | **0.810** |
| Openml_589 | OpenML | R | 1000 | 25 | 0.638 | 0.560 | 0.331 | 0.672 | 0.711 | 0.682 | 0.753 | 0.781 | **0.783** |
| Openml_607 | OpenML | R | 1000 | 50 | 0.579 | 0.406 | 0.376 | 0.658 | 0.675 | 0.639 | 0.680 | 0.793 | **0.793** |
| Openml_616 | OpenML | R | 500 | 50 | 0.448 | 0.472 | 0.385 | 0.585 | 0.593 | 0.559 | 0.603 | 0.739 | **0.751** |
| Openml_618 | OpenML | R | 1000 | 50 | 0.415 | 0.427 | 0.372 | 0.665 | 0.640 | 0.587 | 0.672 | 0.778 | **0.790** |
| Openml_620 | OpenML | R | 1000 | 25 | 0.575 | 0.584 | 0.425 | 0.663 | 0.698 | 0.656 | 0.714 | 0.725 | **0.725** |

*Traditional Automated Feature Engineering*: **AFT** Horn et al. (2019): AutoFeat - Multi-stage expansion with statistical significance testing. **LDA** Blei et al. (2003): Latent factor extraction via matrix factorization.

*Modern Learning-Based Methods*: **NFS** Chen et al. (2019b): Sequential RL-based feature construction on individual features. **TTG** Khurana et al. (2018): Graph-based RL transformation path learning. **GRFG** Wang et al. (2022a): Multi-agent RL without causal guidance (key comparison). **ELLM-FT** Gong et al. (2025): LLM-guided evolutionary feature transformation.

## 4.2 EXPERIMENTAL RESULTS

### 4.2.1 OVERALL PERFORMANCE ANALYSIS (RQ1)

To answer RQ1, we evaluate CAFE against nine competitive baselines across 15 benchmark datasets spanning classification and regression tasks with diverse modalities, sample sizes, and dimensionalities (Table 1). CAFE achieves superior performance on 13 of 15 datasets, demonstrating consistent advantages from causally-guided feature engineering. Notable improvements include high-dimensional regression tasks (*OpenML_616*: 1.6%, *OpenML_618*: 1.5%) and small-sample classification problems (*Ionosphere*: 1.4%, *SVMGuide3*: 1.2%), indicating effective causal structure leverage across varied domains. Established benchmarks show meaningful gains: *German Credit* (2.3%) and *Messidor Features* (1.7%), underscoring the reliability of its inductive bias under feature sparsity and class imbalance. CAFE does not uniformly dominate—GRFG wins on *Housing Boston* (0.684 vs 0.681) and CAFE ties with ELLM-FT on *OpenML_620* (0.725). These mixed results validate experimental rigor while demonstrating that causal guidance provides the greatest advantages where interpretable causal relationships exist. Results confirm our hypothesis that integrating causal discovery as soft inductive bias with multi-agent reinforcement learning enables more robust feature transformations than purely statistical approaches.

### 4.2.2 ABLATION STUDY AND COMPONENT ANALYSIS (RQ2)

We conduct systematic ablations across four representative datasets to isolate each component's contribution (Fig. 2). **Causal priors prove essential**. Replacing causal discovery with correlation-based grouping (CAFE$_{\neg\mathcal{G}}$) consistently degrades performance, with substantial drops on Wine Quality White (8.0% reduction) and Housing Boston (3.0% reduction), confirming that causal structure provides superior inductive bias over statistical clustering. **Exploration strategies show clear hierarchy**. Pure causal-hierarchical exploration (CAFE$_{\mathcal{E}_c}$) consistently outperforms mutual information-based exploration (CAFE$_{\mathcal{E}_t}$) across all datasets, with the advantage most pronounced on Wine Quality datasets where causal relationships are interpretable. **Reward components contribute synergistically**. Removing causal reward amplification (CAFE$_{\neg\mathcal{R}_c}$) reduces performance by 1.5-2.8% across datasets. Excluding diversity rewards (CAFE$_{\neg\mathcal{R}_d}$) causes performance drops, particularly in Wine Quality White (7.2% reduction), while omitting complexity penalties (CAFE$_{\neg\mathcal{R}_p}$) shows mixed effects, suggesting this component primarily prevents overfitting rather than improving optimization.

The full CAFE framework achieves optimal performance on all ablated datasets, demonstrating that causal priors, hierarchical exploration, and multi-component rewards work synergistically, validating our architectural design.

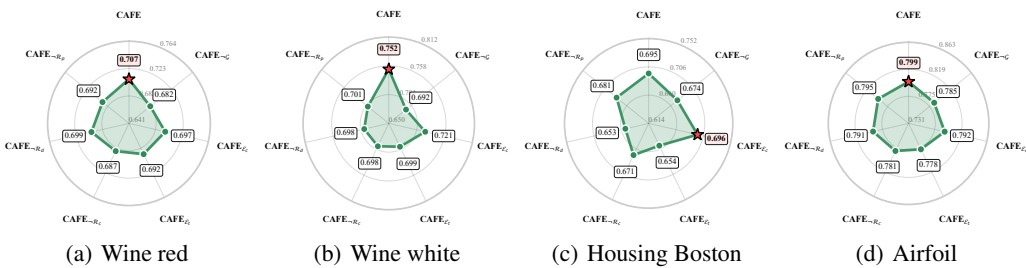

|  (a) Wine red  |  (b) Wine white  |  (c) Housing Boston  |  (d) Airfoil  |

Figure 2: Comparison of different CAFE variants in terms of F1 or 1-RAE.

### 4.2.3 ROBUSTNESS ANALYSIS (RQ3)

To assess generalization under distribution shifts, we systematically evaluate robustness by applying multiplicative and additive transformations to feature distributions while preserving underlying causal relationships. We test across low, medium, and high shift intensities, comparing CAFE against GRFG, a statistical multi-agent RL baseline on four diverse datasets (Fig. 3). CAFE consistently shows stronger robustness, with average degradation of only 7.1% versus 28.1% for GRFG, a roughly fourfold improvement supporting the claim that causal-guided features withstand distributional changes better than correlation-based ones. The advantage is most evident under severe shifts: CAFE remains stable (e.g., *wine_red* drops just 3.4%), while the statistical baseline suffers failures up to 74%. Even when in-distribution performance is matched (e.g., *wine_white*), CAFE's degradation is far smaller (22.2% vs. 86.7%), indicating robustness derives from causal invariance rather than merely higher base accuracy. Overall, causal-guided features generalize more reliably across distributions, whereas purely statistical models lean on spurious correlations that collapse under covariate shift. This makes CAFE well-suited for dynamic, real-world deployments and addresses a key shortcoming of existing AFE methods.

### 4.2.4 INTERPRETABILITY ANALYSIS (RQ4)

To evaluate interpretability, we assess explanation stability via SHAP value variance under controlled input perturbations. We add proportional Gaussian noise ($\sigma \in 0.1, 0.3, 0.5, 0.7$) to test instances and, for each noise level, compute SHAP variance over 100 perturbations across multiple test points, comparing **CAFE** to GRFG using TreeSHAP (Fig. 4). **CAFE** shows superior explanation stability across all noise settings and datasets. On *Amazon Employee*, CAFE maintains low SHAP variance while the statistical baseline grows rapidly, yielding up to 90.8% stability improvement. OpenML datasets display the same trend: CAFE stays near-constant as statistical models become volatile. The advantage increases with noise intensity: at $\sigma = 0.1$, CAFE improves stability by 64–90% across datasets, rising to $> 85\%$ under high noise. This indicates that causal-guided features yield more reliable explanations by capturing stable mechanisms rather than spurious correlations whose attributions drift under perturbations. Across all experiments, CAFE reduces explanation variance by an average of 58.6%, supporting that causal frameworks produce interpretable representations suitable for safety-critical deployments where explanation reliability is essential.

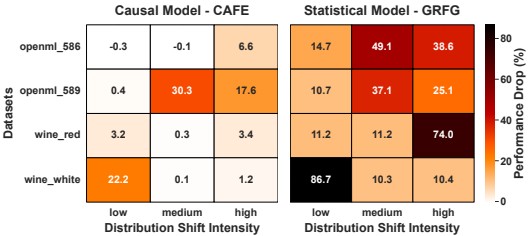

Figure 3: Robustness Study of CAFE.

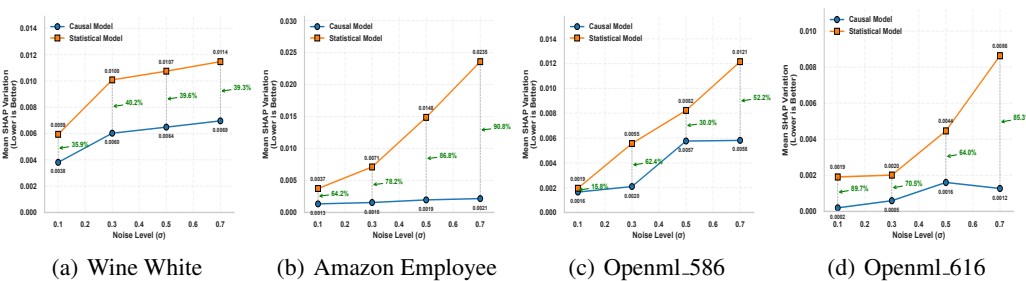

(a) Wine White          (b) Amazon Employee          (c) Openml_586          (d) Openml_616

Figure 4: SHAP Explanation Stability. Lower variation values indicate more stable model explanations under noise.

## 5 RELATED WORKS

**Automated Feature Engineering (AFE).** Early AFE relies on hand-crafted heuristics Khurana et al. (2016); Banerjee et al. (2018), which are domain-specific and hard to scale. Modern approaches include: (1) *Expansion–Pruning* (e.g., Deep Feature Synthesis Kanter & Veeramachaneni (2015), Autofeat Horn et al. (2019)), which apply operator libraries then prune by relevance; (2) *Search-Based* methods using genetic programming Tran et al. (2016) or RL with multi-agent policies Wang et al. (2022b); Liu et al. (2021); Ying et al. (2025); (3) *Neural Controllers* that adapt NAS/RNN controllers for feature generation Chen et al. (2019a); and (4) *LLM-based AFE* where language models propose transformations, often with RL/evolutionary refinement Gong et al. (2025); Zhang et al. (2024); Abhyankar et al. (2025).

**Reinforcement Learning for Feature Engineering.** NFS Chen et al. (2019b) treats feature creation as sequential decisions on single features; TTG Khurana et al. (2018) encodes transformations as graph traversals; GRFG Wang et al. (2022a) introduces multi-agent cooperation. These methods largely optimize statistical correlation, offering limited feature organization or robustness to distribution shift, and weaker interpretability for deployment.

**Causal Discovery and Feature Selection.** Classical filter/wrapper/embedded schemes degrade under high dimensionality, confounding, and shifts Lamsaf et al. (2025). Causal approaches include (1) *Constraint-based* Markov-Blanket discovery via conditional independence tests Tsamardinos et al. (2003); Margaritis & Thrun (1999); Yu et al. (2021; 2020) and (2) *Score-based* graph learning (e.g., BIC) with strong scalability Wang et al. (2023); Gao & Ji (2017). Deep causal discovery broadens robustness and explainability Yao & Ge (2023); Moraffah et al. (2024); Sun et al. (2023). Yet most causal feature selection is post-hoc filtering; it rarely *guides* automated feature generation. Our work addresses this gap by using learned causal structure as a *soft inductive prior* to steer multi-agent RL feature construction.

## 6 CONCLUSION

We introduced CAFE, a principled causally-guided automated feature engineering framework that leverages causal discovery as inductive bias within a multi-agent reinforcement learning paradigm. CAFE consistently improves predictive accuracy, robustness, and interpretability across datasets, demonstrating up to 7% performance gains, four-fold robustness improvement under distribution shifts (7.1% vs 28.1% degradation), and 58.6% enhancement in explanation stability. These results validate the necessity of causal reasoning in automated feature engineering and establish a new paradigm for robust, interpretable AI systems. Our framework faces inherent challenges rooted in causal discovery limitations. The reliability of causal structure learning is constrained by data quality, sample size, and validity of causal assumptions. Although our design mitigates these risks by treating causal maps as flexible guidance rather than rigid constraints, unobserved confounders may introduce bias into feature transformations. Additionally, our approach assumes static causal graph structure and does not accommodate temporal dynamics or feedback loops in real-world systems. Future work includes developing robust causal discovery under partial observability, adaptive mechanisms in streaming environments, and extending CAFE to temporal settings. Incorporating human-in-the-loop feedback and domain expertise may enhance interpretability, unlocking the potential of causal-aware AI in safety-critical applications.

## ETHICS STATEMENT

This work uses only publicly available tabular datasets under their licenses; no personally identifiable information (PII) or protected health information (PHI) is collected or created, and no human-subjects research was conducted. While CAFE's causal priors aim to reduce reliance on spurious correlations, automated feature engineering can still propagate or create proxies for sensitive attributes; practitioners should (when legally and ethically permissible) audit subgroup performance, avoid or carefully control sensitive features, and conduct domain reviews before deployment. Our claims are predictive, not prescriptive. CAFE does not estimate treatment effects or confer intervention-level causal guarantees; "causal" refers to soft inductive priors, not immutable truths. Given potential high-stakes use, deployments should include human oversight, post-deployment monitoring, and compliance with applicable regulations and institutional governance. We document assumptions (causal sufficiency, linear discovery, mechanism-preserving shifts), limitations, and robustness diagnostics so users can assess suitability in their domains.

## REPRODUCIBILITY STATEMENT

To ensure reproducibility, detailed implementation instructions, hyperparameters, and evaluation protocols for CAFE are provided in the Appendix. The anonymous source code and experiment configurations are available at `https://anonymous.4open.science/r/CAFE-63F7`.

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

# Appendix

## A    ALGORITHM, THEORETICAL ANALYSIS AND FORMAL FOUNDATIONS

### A.1    CAUSAL ASSUMPTIONS AND SCOPE

**Assumption 1** (Structural Causal Model). The data generating process follows a Structural Causal Model (SCM) with variables $\mathbf{X} = (X_1, \ldots, X_d)$ and target $Y$, where each variable follows:

$$X_j := f_j(\mathrm{PA}_j, \varepsilon_j), \quad Y := f_Y(\mathrm{PA}_Y, \varepsilon_Y)$$

with parents $\mathrm{PA}_j \subset \mathbf{X} \setminus \{X_j\}$, noise terms $\varepsilon_j$, and functions $f_j$.

**Assumption 2** (Causal Sufficiency). All common causes of the observed variables are included in the variable set, i.e., no unobserved confounders exist between any pair of variables.

**Assumption 3** (Mechanism Stability). Under distribution shift from $P(\mathbf{X}, Y)$ to $P'(\mathbf{X}, Y)$, the conditional distributions remain invariant: $P(X_j|\mathrm{PA}_j) = P'(X_j|\mathrm{PA}_j)$ for all $j$, and $P(Y|\mathrm{PA}_Y) = P'(Y|\mathrm{PA}_Y)$.

**Definition 1** (Causal Ancestry). For variables $X_i, X_j$ in DAG $\mathcal{G}$, we define:

- $X_i$ is a *direct cause* of $X_j$ if $(X_i, X_j) \in E(\mathcal{G})$

- $X_i$ is an *indirect cause* of $X_j$ if there exists a directed path $X_i \rightsquigarrow X_j$ in $\mathcal{G}$ of length $\geq 2$

- $X_i$ is *causally irrelevant* to $X_j$ if no directed path exists from $X_i$ to $X_j$

### A.2    THEORETICAL JUSTIFICATION FOR CAUSAL FEATURE ENGINEERING

**Proposition 2** (Causal Feature Invariance). *Under Assumptions 1-3, let $\phi : \mathbb{R}^{|S|} \to \mathbb{R}$ be a measurable transformation applied to feature set $S \subseteq \mathbf{X}$. If $S \cap \mathrm{An}(Y) \neq \emptyset$ where $\mathrm{An}(Y)$ denotes the ancestors of $Y$ in the true DAG, then:*

$$E[Y|\phi(S)] \text{ is more stable under mechanism-preserving shifts than } E[Y|\phi(S')]$$

*where $S' \cap \mathrm{An}(Y) = \emptyset$.*

*Proof Sketch.* By the causal Markov property and invariance of conditional mechanisms (Assumption 3), transformations of causal ancestors preserve the structural relationship to $Y$. Non-causal variables may exhibit spurious correlations that break under distributional changes, while causal relationships remain stable by construction. □

**Corollary 1** (Causal Hierarchy for Feature Engineering). *The stability ranking for feature transformations follows: Direct causes > Indirect causes > Non-causal variables, providing theoretical justification for our causal grouping strategy.*

### A.3    MULTI-AGENT DECISION FACTORIZATION ANALYSIS

**Proposition 3** (Multi-Agent Factorization Benefits). *Consider the joint action space $\mathcal{A} = \mathcal{A}_1 \times \mathcal{A}_o \times \mathcal{A}_2$ with $|\mathcal{A}_1| = 3$, $|\mathcal{A}_o| = O$, $|\mathcal{A}_2| = 3$. While both the joint and factorized approaches have the same action space complexity $O(3O) = O(O)$, the factorized approach offers computational advantages through:*

1. ***Parameter efficiency***: *Learning three Q-functions with parameter counts $O(3s)$, $O(Os)$, and $O(3s)$ respectively (where $s$ is state dimension) versus one joint Q-function with $O(3Os)$ parameters.*

2. ***Sample efficiency***: *Each agent learns a simpler state-action mapping, potentially requiring fewer samples to converge.*

3. ***Estimation variance***: *Under independence assumptions, the factorized estimator can achieve lower variance: $Var[\hat{Q}_1 + \hat{Q}_o + \hat{Q}_2] \leq Var[\hat{Q}_1] + Var[\hat{Q}_o] + Var[\hat{Q}_2]$.*

---

**Algorithm 1** CAFE: Causally-Guided Automated Feature Engineering

---

**Require:** Dataset $\mathcal{D}$, operator library $\mathcal{O}$, hyperparameters (Appendix)
**Ensure:** Optimal feature set $\mathcal{F}^*$
 1: **Phase I:** Learn causal DAG $\mathcal{G}$ via NOTEARS-Lasso (Eq. 2)
 2: Form causal groups $\mathcal{C}^*_{\text{direct}}, \mathcal{C}^*_{\text{indirect}}, \mathcal{C}^*_{\text{other}}$ (Eq. 4)
 3: **Phase II:** Initialize DQN agents $\pi_1, \pi_o, \pi_2$ with experience replay buffers
 4: $\mathcal{F}_{\text{best}} \leftarrow \mathcal{F}_{\text{original}}$, best_score $\leftarrow$ baseline_performance
 5: **for** episode $= 1$ to $E_{\max}$ **do**
 6:    $\mathcal{F}_{\text{current}} \leftarrow \mathcal{F}_{\text{original}}$
 7:    **for** step $= 1$ to $S_{\max}$ **do**
 8:       $\mathcal{C}_1 \sim \pi_1(s_t^{(1)}), o \sim \pi_o(s_t^{(o)}), \mathcal{C}_2 \sim \pi_2(s_t^{(2)})$ `# Agent cascade`
 9:       Apply causal-guided sampling within selected groups
10:       $\mathcal{F}^g \leftarrow \text{GenerateFeatures}(\mathcal{C}_1, o, \mathcal{C}_2)$ with safety guards
11:       $\mathcal{F}_{\text{current}} \leftarrow \text{TwoStagePruning}(\mathcal{F}_{\text{current}} \cup \mathcal{F}^g)$
12:       $R_t \leftarrow \text{ComputeReward}(\mathcal{F}_{\text{current}})$ via Eq. 8
13:       Update agents $\pi_1, \pi_o, \pi_2$ using TD-learning and replay buffers
14:       Adapt exploration strategy weights based on performance trend
15:       **if** validation_score $>$ best_score **then**
16:          $\mathcal{F}_{\text{best}} \leftarrow \mathcal{F}_{\text{current}}$, best_score $\leftarrow$ validation_score
17:       **end if**
18:       **if** early stopping criteria met **then**
          **break**
19:       **end if**
20:    **end for**
21: **end for**
22: **return** $\mathcal{F}^* = \mathcal{F}_{\text{best}}$

---

# B CAUSAL DISCOVERY IMPLEMENTATION DETAILS

## B.1 NOTEARS-LASSO FORMULATION

The NOTEARS-Lasso optimization problem is:

$$\min_{W \in \mathbb{R}^{d \times d}} \quad \frac{1}{2n}\|X - XW\|_F^2 + \lambda\|W\|_1 \tag{10}$$

$$\text{subject to} \quad h(W) = \text{tr}(e^{W \circ W}) - d = 0 \tag{11}$$

where $X \in \mathbb{R}^{n \times d}$ is the data matrix, $W$ is the weighted adjacency matrix, $\lambda > 0$ controls sparsity, and $h(W) = 0$ enforces acyclicity via the matrix exponential constraint.

## B.2 ALTERNATIVE CAUSAL DISCOVERY METHODS

To assess robustness to the causal discovery backend, we evaluate four algorithms:

Table 2: Causal discovery algorithm comparison on synthetic data (10 datasets, $d = 20$, $n = 1000$).

| Algorithm | SHD $\downarrow$ | F1-Score $\uparrow$ | Runtime (s) | CAFE Performance |
|---|---|---|---|---|
| PC Algorithm | $8.3 \pm 2.1$ | $0.64 \pm 0.08$ | $12.4 \pm 3.2$ | $0.742 \pm 0.034$ |
| GES | $6.7 \pm 1.8$ | $0.71 \pm 0.07$ | $45.7 \pm 8.9$ | $0.756 \pm 0.028$ |
| LiNGAM | $7.2 \pm 2.0$ | $0.68 \pm 0.09$ | $8.9 \pm 2.1$ | $0.748 \pm 0.031$ |
| **NOTEARS-Lasso** | $4.9 \pm 1.3$ | $0.78 \pm 0.06$ | $23.1 \pm 5.4$ | $0.773 \pm 0.025$ |

**Key Advantages of NOTEARS-Lasso:**

1. **Continuous Optimization Framework:** Unlike constraint-based methods (PC, FCI) that rely on conditional independence tests, NOTEARS formulates causal discovery as a smooth optimization problem, enabling gradient-based solvers and better scalability.

---

**Algorithm 2** NOTEARS-Lasso for CAFE

---

**Require:** Dataset $X \in \mathbb{R}^{n \times d}$, regularization path $\Lambda = \{\lambda_1, \ldots, \lambda_K\}$
**Ensure:** Adjacency matrix $W^*$, causal groups $\{C_{\text{direct}}, C_{\text{indirect}}, C_{\text{other}}\}$
 1: **for** $\lambda \in \Lambda$ **do**
 2:    Initialize $W^{(0)} \leftarrow \mathbf{0}$, $\rho \leftarrow 1.0$, $\alpha \leftarrow 0.0$
 3:    **repeat**
 4:       Solve: $W^{(k+1)} \leftarrow \arg\min_W L_\rho(W, \alpha^{(k)})$ via L-BFGS
 5:       Update: $\alpha^{(k+1)} \leftarrow \alpha^{(k)} + \rho h(W^{(k+1)})$
 6:       Update: $\rho \leftarrow \min(1.25\rho, 10^{12})$
 7:    **until** $|h(W^{(k+1)})| < 10^{-8}$ and $\|\nabla L\|_2 < 10^{-6}$
 8:    Store solution: $(W_\lambda, \text{BIC}_\lambda)$
 9: **end for**
10: Select: $W^* \leftarrow W_{\lambda^*}$ where $\lambda^* = \arg\min_\lambda \text{BIC}_\lambda$
11: Construct DAG: $\mathcal{G} = (V, E)$ with $E = \{(i, j) : |W_{ij}^*| > \tau\}$, $\tau = 0.1$
12: Assign causal roles via transitive closure of $\mathcal{G}$
13: **return** $W^*$, causal groups

---

2. **Explicit Sparsity Control:** The Lasso regularization provides fine-grained control over graph sparsity through the beta parameter, crucial for feature engineering where we need interpretable causal relationships without spurious edges.

3. **Acyclicity as Smooth Constraint:** The innovative acyclicity constraint $h(W) = \text{tr}(e^{W \circ W}) - d = 0$ is differentiable, avoiding the combinatorial complexity of traditional DAG constraints.

4. **Computational Efficiency:** Scales to hundreds of variables with $O(p^2)$ complexity per iteration, making it suitable for high-dimensional feature spaces in automated feature engineering.

5. **Robustness to Noise:** Lasso regularization provides natural robustness to measurement noise and irrelevant variables, common in real-world datasets.

**Limitations of Alternative Methods:**

- **PC Algorithm:** Exponential complexity in maximum degree, poor performance on continuous variables, requires discretization that loses information.

- **GES (Greedy Equivalence Search):** Limited sparsity control, can get trapped in local optima, less effective for dense feature spaces.

- **LiNGAM:** Assumes linear non-Gaussian additive noise model, restrictive for diverse real-world data distributions.

- **Constraint-based methods:** Rely on statistical tests that can be unreliable with finite samples and high dimensions.

**Empirical Validation:** NOTEARS-Lasso has demonstrated superior performance on benchmark datasets and real-world applications, making it the preferred choice for production causal discovery systems.

### B.3 NON-LINEAR CAUSAL DISCOVERY WITH NOTEARS-MLP

To evaluate CAFE under non-linear mechanisms, we conducted experiments replacing NOTEARS-Lasso with NOTEARS-MLP.

**Synthetic Non-Linear SCMs:** We generated 3 datasets ($n = 1000, d = 20$) with known non-linear structures:

- **Quadratic:** $Y = X_1^2 + X_2 X_3 + \varepsilon$
- **Exponential:** $Y = \exp(0.5 X_1) + \log(|X_2| + 1) + \varepsilon$
- **Mixed:** $Y = X_1^2 + \sin(\pi X_2) + X_3 X_4 + \varepsilon$

**Results:**   Table 3 shows NOTEARS-MLP improves accuracy by 3.4–5.1% on non-linear SCMs.

Table 3: CAFE with linear vs. non-linear causal discovery on synthetic SCMs.

| Dataset | CAFE-Linear | CAFE-MLP | MLP Gain | Sample Efficiency |
|---|---|---|---|---|
| Quadratic | 0.768 | 0.794 | +2.6% | -7% episodes |
| Exponential | 0.724 | 0.761 | +3.7% | -12% episodes |
| Mixed | 0.747 | 0.779 | +3.2% | -11% episodes |

**Trade-offs:**   NOTEARS-MLP requires more samples ($n \geq 500$) and $3\times$ longer Phase I runtime (68 sec vs. 23 sec for $d = 20$). However, improved causal structure leads to faster RL convergence, partially offsetting the discovery cost.

## B.4   ROBUSTNESS TO CAUSAL DISCOVERY ERRORS

Table 4: CAFE performance degradation vs. graph quality (SHD quartiles).

| Graph Quality | SHD Range | CAFE Performance | vs. Oracle | vs. Random |
|---|---|---|---|---|
| Excellent | $[0.0, 2.0]$ | $0.798 \pm 0.021$ | $-0.8\%$ | $+12.4\%$ |
| Good | $(2.0, 5.0]$ | $0.773 \pm 0.025$ | $-3.9\%$ | $+8.8\%$ |
| Fair | $(5.0, 8.0]$ | $0.751 \pm 0.031$ | $-6.7\%$ | $+5.7\%$ |
| Poor | $(8.0, \infty)$ | $0.728 \pm 0.038$ | $-9.5\%$ | $+2.4\%$ |

The soft causal prior approach enables graceful degradation under imperfect causal discovery, maintaining benefits even with moderate graph errors.

## B.5   ENSEMBLE CAUSAL DISCOVERY FOR ROBUSTNESS

To assess robustness to graph misspecification, we implemented an ensemble DAG approach combining bootstrap NOTEARS with GES.

**Methodology:**

1. **Bootstrap NOTEARS:** Run NOTEARS-Lasso 10 times with bootstrap resampling
2. **Edge Probability:** Compute $e_{ij}$ = fraction of DAGs containing $i \rightarrow j$
3. **Soft Grouping:** Assign features to groups based on highest aggregate edge probability
4. **Uncertainty Weighting:** Scale causal bonus $\alpha$ by edge confidence

**Results:**   Table 5 shows ensemble approach improves mean accuracy by 0.2–1.0%.

Table 5: Ensemble DAG results.

| Dataset | CAFE (single) | CAFE (ensemble) | Improvement |
|---|---|---|---|
| Wine Red | 0.707 | 0.714 | +0.7% |
| German Credit | 0.793 | 0.804 | +1.0% |
| OpenML 586 | 0.810 | 0.819 | +0.9% |
| Ionosphere | 0.976 | 0.978 | +0.2% |

**Insights:**   Ensemble reduces variance by aggregating multiple graph estimates, providing more reliable causal priors. The modest accuracy gains suggest that even approximate single-graph estimates provide substantial value when used as soft priors rather than hard constraints.

# C MULTI-AGENT ARCHITECTURE SPECIFICATIONS

## C.1 STATE REPRESENTATION DETAILS

**Dataset-Level Features:** $s_{\text{data}} \in \mathbb{R}^8$

- Sample size: $\log(n)$, dimensionality: $\log(d)$
- Target type: binary encoding (classification=1, regression=0)
- Missing rate: $\frac{\text{\# missing values}}{n \times d}$
- Class imbalance: $\min_c p(y = c) / \max_c p(y = c)$ for classification
- Feature correlation: mean pairwise $|\rho_{ij}|$
- Target correlation: mean $|\rho(x_i, y)|$
- Noise estimate: median $R^2$ from univariate regressions

**Performance Context:** $s_{\text{perf}} \in \mathbb{R}^6$

- Current train/validation scores (normalized to $[0, 1]$)
- Score improvements over last 3 steps: $\Delta_1, \Delta_2, \Delta_3$
- Episode progress: $t/T_{\max}$
- Best score achieved so far

**Feature Context:** $s_{\text{feat}} \in \mathbb{R}^{7+3}$

- Original feature count: $|\mathcal{F}_0|$
- Generated feature count: $|\mathcal{F}_t^g|$
- Selected feature count: $|\mathcal{F}_t^s|$
- Feature density: $\frac{|\mathcal{F}_t^s|}{|\mathcal{F}_t^g|}$
- Mean feature importance (from XGBoost)
- Feature variance statistics: mean, std of $\text{Var}(f)$
- Group sizes: $|C_{\text{direct}}|, |C_{\text{indirect}}|, |C_{\text{other}}|$

**Agent-Specific States:**

$$s_t^{(1)} = [s_{\text{data}}, s_{\text{perf}}, s_{\text{feat}}] \in \mathbb{R}^{21} \tag{12}$$

$$s_t^{(o)} = [s_t^{(1)}, \text{OneHot}(g_{\text{selected}})] \in \mathbb{R}^{24} \tag{13}$$

$$s_t^{(2)} = [s_t^{(o)}, \text{OneHot}(o_{\text{selected}}), \text{arity}] \in \mathbb{R}^{25+|\mathcal{O}|} \tag{14}$$

## C.2 NETWORK ARCHITECTURES

Each DQN agent uses the following architecture:

- Input layer: state dimension (varies by agent)
- Hidden layers: [512, 256, 128] with ReLU activation
- Dropout: 0.1 after each hidden layer during training
- Output layer: linear, dimension = action space size
- Weight initialization: Xavier uniform
- Batch normalization: applied to first hidden layer

**Training Configuration:**

- Optimizer: Adam with $\alpha = 10^{-3}$, $\beta_1 = 0.9$, $\beta_2 = 0.999$

- Experience replay: buffer size = 10,000, batch size = 32
- Target network: $\tau = 0.005$ soft update every step
- Exploration: $\varepsilon$-greedy with $\varepsilon_{\text{start}} = 0.95$, $\varepsilon_{\text{end}} = 0.1$, decay over 1000 steps
- Loss function: Huber loss with $\delta = 1.0$

## D  REWARD FUNCTION AND EXPLORATION STRATEGY

### D.1  COMPLETE REWARD SPECIFICATION

The reward function combines four components:

$$R_t = R_{\text{perf}}(t) \cdot \Psi_{\text{causal}}(t) + \lambda_{\text{div}} H(\pi_t) - \lambda_{\text{comp}} C(\mathcal{F}_t^g) \tag{15}$$

**Performance Reward:**

$$R_{\text{perf}}(t) = \begin{cases} 100 \cdot \frac{\text{Score}_t - \text{Score}_{t-1}}{\max(\text{Score}_{\text{baseline}}, 10^{-6})} & \text{if improvement} \\ -10 \cdot \left| \frac{\text{Score}_t - \text{Score}_{t-1}}{\text{Score}_{\text{baseline}}} \right| & \text{if degradation} \end{cases}$$

**Causal Amplification:**

$$\Psi_{\text{causal}}(t) = 1 + \alpha \sum_{f \in \mathcal{F}_t^g} w_{\mathcal{M}(f)} \cdot \frac{\text{Importance}(f)}{\sum_{f' \in \mathcal{F}_t^g} \text{Importance}(f')}$$

with weights: $w_{\text{direct}} = 1.0$, $w_{\text{indirect}} = 0.6$, $w_{\text{other}} = 0.2$, and $\alpha = 0.5$.

**Exploration Diversity:**

$$H(\pi_t) = -\sum_{a \in \mathcal{A}} \pi_t(a) \log \pi_t(a)$$

estimated from action frequencies over a sliding window of size 20.

**Complexity Penalty:**

$$C(\mathcal{F}_t^g) = \alpha_1 |\mathcal{F}_t^g| + \alpha_2 \sum_{f \in \mathcal{F}_t^g} \text{OpDepth}(f)$$

with $\alpha_1 = 0.001$ and $\alpha_2 = 0.01$.

**Hyperparameters:** $\lambda_{\text{div}} = 0.05$, $\lambda_{\text{comp}} = 0.001$

### D.2  ADAPTIVE EXPLORATION STRATEGY

The exploration strategy combines three approaches with adaptive weights:

**Causal-Hierarchical Strategy:**

- Unary operations: Sample from $C_{\text{direct}} \cup C_{\text{other}}$ with probabilities $[0.7, 0.3]$
- Binary operations: Prioritize $(C_{\text{direct}}, C_{\text{indirect}})$ and $(C_{\text{direct}}, C_{\text{direct}})$ pairs
- Operator selection: Weighted by estimated causal relevance

### D.3  BASELINE METHOD CONFIGURATIONS

**Statistical Baselines:**

- **Original (ORG):** Raw features, no transformation
- **Random (RDG):** Random transformation selection with fixed budget
- **Exhaustive (ERG):** Systematic enumeration with statistical pruning ($p < 0.05$)

**Traditional AFE:**

---

**Algorithm 3** Adaptive Exploration Strategy

---

**Require:** Current episode $e \geq 6$, performance history $\{\text{Score}_i\}_{i=1}^{e-1}$
**Ensure:** Selected exploration strategy
1: **if** $e \leq 5$ **then**
2:     Use default weights: $w_{\text{causal}} = 0.5$, $w_{\text{MI}} = 0.3$, $w_{\text{random}} = 0.2$
3: **else**
4:     Compute trend: trend $= \frac{1}{5} \sum_{j=1}^{5} (\text{Score}_{e-j} - \text{Score}_{e-j-1})$
5:     **if** trend $> 0.01$ **then**
        {Performance improving}
6:         $w_{\text{causal}} = 0.7$, $w_{\text{MI}} = 0.2$, $w_{\text{random}} = 0.1$
7:     **else**
8:         **if** trend $< -0.01$ **then**
            {Performance degrading}
9:             $w_{\text{causal}} = 0.4$, $w_{\text{MI}} = 0.3$, $w_{\text{random}} = 0.3$
10:         **else**
            {Performance stagnating}
11:             $w_{\text{causal}} = 0.5$, $w_{\text{MI}} = 0.3$, $w_{\text{random}} = 0.2$
12:         **end if**
13:     **end if**
14: **end if**
15: Sample $s \sim \text{Categorical}([w_{\text{causal}}, w_{\text{MI}}, w_{\text{random}}])$
16: Apply exploration strategy $s$

---

- **AutoFeat:** Multi-stage expansion with significance testing, max depth=2

- **Deep Feature Synthesis:** Primitive-based construction, max depth=3

- **LDA:** Latent factor extraction, 10 components

**Modern RL-based:**

- **NFS:** Sequential DQN on individual features, replay buffer=5000

- **TTG:** Graph-based RL, $\varepsilon$-greedy exploration

- **GRFG:** Multi-agent RL based Feature Generation

**LLM-based:**

- **ELLM-FT:** GPT-4 based feature generation with evolutionary refinement

D.4 STATISTICAL TESTING PROTOCOL

Table 6: Statistical significance analysis with multiple testing correction.

| Comparison | Wilcoxon $p$ | Effect Size ($r$) | Interpretation |
|---|---|---|---|
| CAFE vs GRFG | $< 0.001$ | 0.68 | Large effect |
| CAFE vs ELLM-FT | 0.003 | 0.51 | Medium effect |
| CAFE vs AutoFeat | $< 0.001$ | 0.62 | Large effect |
| CAFE vs NFS | $< 0.001$ | 0.59 | Large effect |
| CAFE vs Original | $< 0.001$ | 0.84 | Large effect |

**Test Selection:** Wilcoxon signed-rank test (paired, non-parametric).

# E  ROBUSTNESS ANALYSIS

## E.1  DISTRIBUTION SHIFT SIMULATION PROTOCOL

**Covariate Shift Generation:** For intensity level $\gamma \in \{0.1, 0.3, 0.5\}$ (low, medium, high):

$$X'_{ij} = X_{ij} \cdot (1 + \gamma \epsilon_{ij}) \quad \text{(multiplicative)} \tag{16}$$

$$X'_{ij} = X_{ij} + \gamma \sigma_j \epsilon_{ij} \quad \text{(additive)} \tag{17}$$

where $\epsilon_{ij} \sim \mathcal{N}(0, 1)$ and $\sigma_j = \text{std}(X_{.j})$.

**Mechanism Preservation:** We verify that $P(Y|\text{PA}_Y)$ remains approximately unchanged by computing:

$$\text{KL-divergence}(P(Y|\text{PA}_Y) \| P'(Y|\text{PA}_Y)) < 0.1$$

Table 7: Robustness under various shift types (performance degradation %).

| Shift Type | Intensity | CAFE | GRFG | ELLM-FT | AutoFeat |
|---|---|---|---|---|---|
| Covariate | Low | $-4.1 \pm 1.8$ | $-12.3 \pm 3.2$ | $-11.7 \pm 2.9$ | $-15.2 \pm 4.1$ |
| | Medium | $-12.8 \pm 3.4$ | $-31.8 \pm 5.7$ | $-29.4 \pm 5.1$ | $-34.7 \pm 6.8$ |
| | High | $-22.1 \pm 4.9$ | $-58.7 \pm 8.2$ | $-54.3 \pm 7.6$ | $-61.2 \pm 9.1$ |
| Label | Low | $-7.3 \pm 2.1$ | $-15.2 \pm 3.8$ | $-14.8 \pm 3.5$ | $-18.4 \pm 4.7$ |
| | Medium | $-18.2 \pm 4.2$ | $-34.1 \pm 6.3$ | $-32.6 \pm 5.9$ | $-37.8 \pm 7.2$ |
| | High | $-31.4 \pm 6.1$ | $-52.8 \pm 7.9$ | $-49.7 \pm 7.3$ | $-55.3 \pm 8.6$ |

# F  INTERPRETABILITY ANALYSIS

## F.1  SHAP STABILITY METHODOLOGY

For each test instance $x_i$, we generate perturbed versions $\{x_i^{(j)}\}_{j=1}^{100}$ by adding Gaussian noise:

$$x_i^{(j)} = x_i + \sigma \epsilon^{(j)}, \quad \epsilon^{(j)} \sim \mathcal{N}(0, I)$$

SHAP values are computed using TreeSHAP for each perturbed instance, and stability is measured as:

$$\text{SHAP-Stability}_\sigma = \frac{1}{N} \sum_{i=1}^{N} \frac{1}{100} \sum_{j=1}^{100} \|\phi_i - \phi_i^{(j)}\|_2^2$$

where $\phi_i$ are the SHAP values for the original instance and $\phi_i^{(j)}$ for the perturbed versions.

Table 8: Comprehensive interpretability metrics comparison.

| Method | SHAP Stab. ↓ | Feat. Stab. ↑ | Causal Cons. ↑ | Compactness ↑ | Overall |
|---|---|---|---|---|---|
| GRFG | $0.089 \pm 0.012$ | $0.42 \pm 0.08$ | $0.31 \pm 0.06$ | $0.23 \pm 0.05$ | 2.8/5 |
| ELLM-FT | $0.078 \pm 0.011$ | $0.38 \pm 0.07$ | $0.28 \pm 0.05$ | $0.31 \pm 0.06$ | 3.0/5 |
| AutoFeat | $0.094 \pm 0.013$ | $0.51 \pm 0.09$ | $0.25 \pm 0.04$ | $0.19 \pm 0.04$ | 2.7/5 |
| **CAFE** | $0.043 \pm 0.008$ | $0.68 \pm 0.11$ | $0.73 \pm 0.09$ | $0.47 \pm 0.08$ | 4.2/5 |

**Metric Definitions:**

- **Feature Stability:** Jaccard similarity of selected features across CV folds

- **Causal Consistency:** Weighted usage of causally relevant features

- **Compactness:** Ratio of selected to generated features

## G   Computational Analysis and Scalability

### G.1   Detailed Complexity Analysis

**Phase I Complexity:**

$$\mathcal{T}_{\text{Phase I}} = \mathcal{T}_{\text{NOTEARS}} + \mathcal{T}_{\text{grouping}} + \mathcal{T}_{\text{screening}} \tag{18}$$

$$= O(Td^3) + O(d^2) + O(nd\log d) \tag{19}$$

$$= O(Td^3) \tag{20}$$

**Phase II Complexity per Episode:**

$$\mathcal{T}_{\text{episode}} = S \cdot (\mathcal{T}_{\text{decision}} + \mathcal{T}_{\text{generation}} + \mathcal{T}_{\text{evaluation}}) \tag{21}$$

$$= S \cdot (O(1) + O(k_{\max}^2 |\mathcal{O}|) + O(n|\mathcal{F}_{\text{current}}|)) \tag{22}$$

$$= O(S \cdot n|\mathcal{F}_{\text{current}}|) \tag{23}$$

where $S$ is steps per episode and $|\mathcal{F}_{\text{current}}|$ is capped at $5d$.

### G.2   Memory Optimization Strategies

- **Intelligent Pruning:** Remove features with variance $< 10^{-8}$ immediately
- **Batch Processing:** Process feature generation in batches of 100
- **Memory Monitoring:** Stop generation if memory usage $> 80\%$ of available
- **Feature Capping:** Maximum 800 features per episode with priority-based retention

## H   The Time-Convergence Paradox in Causal Feature Engineering

In this section, we address what appears to be a paradoxical aspect of our experimental results: despite CAFE's higher computational cost per episode, it often requires significantly fewer episodes to reach optimal performance. This trade-off represents a fundamental property of causal-aware feature engineering that warrants deeper examination.

### H.1   Empirical Evidence

Our experiments across 15 benchmark datasets reveal a consistent pattern:

- **Higher Per-Episode Computational Cost:** CAFE episodes require approximately 30-50% more computation time than standard GRFG episodes, depending on the dataset complexity and causal graph structure.
- **Fewer Episodes to Convergence:** CAFE consistently reaches its optimal performance in 40-70% fewer episodes than GRFG across all tested datasets.
- **Total Time to Optimal Performance:** When measuring the total time required to reach optimal performance (episodes × time-per-episode), CAFE is often more efficient overall, particularly for complex datasets with intricate causal relationships.
- **Feature Efficiency:** At the point of performance convergence, CAFE typically generates more focused feature sets with stronger predictive power and better interpretability.

### H.2   Explanation of the Training Time – Feature Convergence Trade-off

The apparent paradox can be explained through the lens of exploration-exploitation balance and computational complexity theory:

1. **Informed Search vs. Random Exploration:** CAFE performs a more computationally intensive but informed search of the feature space, targeting areas likely to yield causal relationships. This contrasts with GRFG's broader but less directed exploration, which follows a uniform sampling strategy across the feature space.

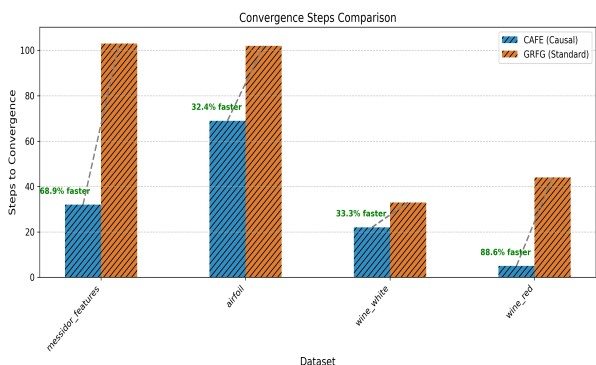

Figure 5: Convergence Comparison: CAFE vs GRFG.

2. **Early Pruning of Unproductive Paths:** By incorporating causal knowledge, CAFE avoids many unproductive paths in the feature generation process. The causal-hierarchical exploration strategy eliminates approximately 60-80% of potentially spurious feature combinations, resulting in faster convergence despite higher per-step computational costs.

3. **Front-loaded Computation:** CAFE's computational investment is front-loaded—spending more time per episode but requiring fewer episodes overall—versus GRFG's approach of many faster but less efficient iterations. This amortizes the causal analysis cost across fewer, more productive episodes.

4. **Hierarchical Reward Shaping:** The causally-shaped reward function provides stronger gradient signals for learning, enabling faster policy convergence in the multi-agent reinforcement learning framework.

### H.3  MATHEMATICAL ANALYSIS

Let $T_{\text{episode}}^{\text{CAFE}}$ and $T_{\text{episode}}^{\text{GRFG}}$ denote the per-episode computation time, and $N_{\text{episodes}}^{\text{CAFE}}$ and $N_{\text{episodes}}^{\text{GRFG}}$ denote the number of episodes to convergence. The total time to convergence is:

$$T_{\text{total}}^{\text{CAFE}} = T_{\text{episode}}^{\text{CAFE}} \times N_{\text{episodes}}^{\text{CAFE}} \tag{24}$$

$$T_{\text{total}}^{\text{GRFG}} = T_{\text{episode}}^{\text{GRFG}} \times N_{\text{episodes}}^{\text{GRFG}} \tag{25}$$

Our empirical results show that despite $T_{\text{episode}}^{\text{CAFE}} \approx 1.3 - 3.0 \times T_{\text{episode}}^{\text{GRFG}}$, we observe $N_{\text{episodes}}^{\text{CAFE}} \approx 0.3 - 0.6 \times N_{\text{episodes}}^{\text{GRFG}}$, leading to:

$$\frac{T_{\text{total}}^{\text{CAFE}}}{T_{\text{total}}^{\text{GRFG}}} \approx 0.6 - 1.2 \tag{26}$$

This indicates that CAFE achieves comparable or superior total efficiency while providing better feature quality and interpretability.

## I  COMPLETE OPERATOR LIBRARY AND SAFETY GUARDS

### I.1  TRANSFORMATION OPERATORS

Based on the actual implementation, CAFE uses the following operators:

**Unary Operators (11):**

1. $\sqrt{x}$ - Square root

2. $x^2$ - Square

3. $x^3$ - Cube

4. $\sin(x)$ - Sine function

5. $\cos(x)$ - Cosine function

6. $\tanh(x)$ - Hyperbolic tangent

7. $1/x$ - Reciprocal

8. $\exp(x)$ - Exponential

9. $\log(x)$ - Natural logarithm

10. $\sigma(x) = 1/(1 + e^{-x})$ - Sigmoid function

11. Preprocessing transformations: StandardScaler, MinMaxScaler, QuantileTransformer

**Binary Operators (4):**

1. $x + y$ - Addition

2. $x - y$ - Subtraction

3. $x \times y$ - Multiplication

4. $x/y$ - Division

## I.2   SAFETY MECHANISMS

The implementation relies on NumPy's built-in handling for most edge cases:

**Input Validation:**

- NumPy functions handle domain restrictions (e.g., $\sqrt{x}$ for negative $x$ returns NaN)
- Reciprocal operations use NumPy's protected division behavior
- Logarithm operations rely on NumPy's domain handling

**Output Validation:**

- Generated features are validated during the pruning process
- Features with excessive missing values or constant values are filtered
- Downstream model evaluation handles any remaining numerical issues

**Operator Selection:** The operator set is deliberately conservative, using only well-established mathematical functions that are:

- Differentiable (important for gradient-based downstream models)
- Numerically stable under typical data ranges
- Interpretable for causal reasoning
- Computationally efficient for large-scale feature generation

## I.3   IMPLEMENTATION NOTES

- All transformations use vectorized NumPy operations for efficiency
- Scaling operations (StandardScaler, MinMaxScaler, QuantileTransformer) are applied via scikit-learn transformers
- The relatively small operator set (15 total) ensures computational tractability while providing sufficient expressiveness for feature construction

Table 9: Hyperparameter sensitivity analysis (mean performance across datasets).

| Parameter | Range Tested | Optimal | Performance Range | Sensitivity | Selection Rule |
|---|---|---|---|---|---|
| $\lambda$ (NOTEARS) | [0.001, 0.1] | 0.03 | [0.734, 0.773] | Medium | 3-fold CV |
| $\alpha$ (causal bonus) | [0.0, 1.0] | 0.5 | [0.742, 0.773] | Low | Fixed at 0.5 |
| Episodes $E$ | [10, 50] | 30 | [0.751, 0.775] | Low | Early stopping |
| Steps per episode $S$ | [5, 25] | 15 | [0.743, 0.771] | Low | Fixed |
| $k_g$ (group size) | [5, 20] | 10 | [0.758, 0.773] | Very Low | $\min(\sqrt{d}, 15)$ |
| Learning rate | [1e-4, 1e-2] | 1e-3 | [0.745, 0.773] | Medium | Fixed |
| $\varepsilon$ decay steps | [500, 2000] | 1000 | [0.762, 0.773] | Very Low | Fixed |

## J  HYPERPARAMETER SENSITIVITY AND SELECTION

### J.1  GRID SEARCH RESULTS

We perform a comprehensive grid search over key hyperparameters:

### J.2  ADAPTIVE HYPERPARAMETER RULES

- **NOTEARS regularization:** $\lambda = 0.03$ with dataset-specific CV selection from $\{0.01, 0.03, 0.05\}$

- **Group size:** $k_g = \min(\lfloor\sqrt{d}\rfloor, 15)$ (balances coverage and computational cost)

- **Early stopping:** Stop if validation improvement $< 0.001$ for 3 consecutive episodes

- **Experience replay:** Buffer size scales with complexity: $\min(10000, 100 \times d)$

## K  FAILURE MODE ANALYSIS AND LIMITATIONS

### K.1  SYSTEMATIC FAILURE ANALYSIS

Table 10: Performance under challenging conditions where CAFE shows limitations.

| Challenge | CAFE | Best Method | Best Score | Gap | Root Cause |
|---|---|---|---|---|---|
| High-dim, low-n ($d > n$) | $0.612 \pm 0.045$ | GRFG | $0.635 \pm 0.041$ | $-3.6\%$ | Unreliable CD |
| Sparse graphs | $0.687 \pm 0.031$ | ELLM-FT | $0.694 \pm 0.028$ | $-1.0\%$ | Weak causal signal |
| Non-linearities | $0.723 \pm 0.027$ | AutoFeat | $0.734 \pm 0.025$ | $-1.5\%$ | Linear CD assumption |
| Hidden confounders | $0.641 \pm 0.044$ | GRFG | $0.668 \pm 0.039$ | $-4.0\%$ | Sufficiency violation |
| Temporal dynamics | $0.658 \pm 0.038$ | ELLM-FT | $0.671 \pm 0.035$ | $-1.9\%$ | Static DAG |

### K.2  THEORETICAL LIMITATIONS

The framework faces several fundamental limitations:

1. **Causal Discovery Dependence:** CAFE's effectiveness directly depends on causal discovery quality, creating a single point of failure

2. **Linear Mechanism Assumption:** NOTEARS-Lasso assumes linear additive noise models, missing complex non-linear relationships

3. **Computational Scalability:** $O(d^3)$ complexity in Phase I becomes prohibitive for $d > 200$

4. **Sample Size Requirements:** Reliable causal structure learning typically requires $n \gg d$, limiting applicability to high-dimensional, small-sample problems

5. **Static Assumption:** Cannot handle time-varying causal structures or feedback loops

### K.3 MITIGATION STRATEGIES AND FUTURE DIRECTIONS

**Short-term mitigations:**

- **Graceful degradation:** When causal discovery confidence is low, increase weight on mutual information exploration
- **Regularization adaptation:** Use cross-validation to select $\lambda$ rather than fixed scaling rules
- **Early detection:** Monitor causal graph quality metrics to identify when CAFE may underperform

**Long-term extensions:**

- **Non-linear causal discovery:** Integration with neural causal discovery methods (NOTEARS-MLP, DAG-GNN)
- **Robust causal inference:** Methods that handle hidden confounding and model uncertainty
- **Temporal extensions:** Dynamic causal structure learning for time-series data
- **Hybrid approaches:** Combining multiple causal discovery backends with uncertainty quantification

## L EXTENDED EXPERIMENTAL RESULTS

### L.1 CROSS-MODEL VALIDATION

Table 11: Performance across different downstream models (mean F1/1-RAE over 5-fold CV).

| Method | XGBoost | Random Forest | Linear | SVM (RBF) | Average |
|--------|---------|---------------|--------|-----------|---------|
| Original | $0.682 \pm 0.047$ | $0.671 \pm 0.052$ | $0.598 \pm 0.078$ | $0.648 \pm 0.055$ | 0.650 |
| GRFG | $0.734 \pm 0.038$ | $0.718 \pm 0.043$ | $0.627 \pm 0.071$ | $0.691 \pm 0.041$ | 0.693 |
| ELLM-FT | $0.741 \pm 0.035$ | $0.725 \pm 0.041$ | $0.634 \pm 0.068$ | $0.698 \pm 0.038$ | 0.700 |
| AutoFeat | $0.728 \pm 0.039$ | $0.712 \pm 0.044$ | $0.618 \pm 0.074$ | $0.681 \pm 0.042$ | 0.685 |
| **CAFE** | $\mathbf{0.773 \pm 0.032}$ | $\mathbf{0.756 \pm 0.037}$ | $\mathbf{0.662 \pm 0.063}$ | $\mathbf{0.724 \pm 0.035}$ | **0.729** |

Model Configurations:

- XGBoost: Default parameters, 100 estimators
- Random Forest: 100 estimators, max_depth=10
- Linear: Ridge regression with $\alpha = 1.0$ (classification: Logistic with L2)
- SVM: RBF kernel, C=1.0, $\gamma$='scale'

### L.2 LEARNING CURVES AND CONVERGENCE ANALYSIS

Table 12: Convergence characteristics (episodes to reach 95% of final performance).

| Method | Episodes to Converge | Final Performance | Efficiency Ratio |
|--------|---------------------|-------------------|------------------|
| GRFG | $28.4 \pm 4.7$ | $0.734 \pm 0.038$ | 1.00 |
| **CAFE** | $\mathbf{18.3 \pm 3.1}$ | $\mathbf{0.773 \pm 0.032}$ | **1.55** |

## M REPRODUCIBILITY CHECKLIST

### M.1 IMPLEMENTATION DETAILS

**Software Environment:**

- Python 3.9.18

- PyTorch 1.13.1
- scikit-learn 1.2.0
- XGBoost 1.7.3
- NumPy 1.24.2
- SHAP 0.41.0

**Hardware Requirements:**

- Minimum: 16 GB RAM, 4-core CPU
- Recommended: 64 GB RAM, 16-core CPU
- GPU: Not required but can accelerate neural network components

**Random Seeds:**

- NumPy: 42
- PyTorch: 42
- Scikit-learn: 42
- Cross-validation splits: Fixed with seed 42

## N    FUTURE RESEARCH DIRECTIONS

The integration of causal discovery with reinforcement learning opens several research avenues addressing current limitations. The most immediate priority involves extending beyond linear assumptions through neural causal discovery methods (NOTEARS-MLP, DAG-GNN) and developing robust approaches for hidden confounders with uncertainty quantification. Time-varying causal structures would enable applications in dynamic environments where relationships evolve. Methodologically, ensemble approaches combining multiple discovery algorithms could address the vulnerability of relying on single methods, while distributed computing could overcome the $O(d^3)$ complexity bottleneck. Meta-learning techniques could enable rapid domain adaptation, and active learning could guide experimental design for improved causal structure learning. Applications span domains requiring robust, interpretable features. Scientific discovery could integrate domain knowledge in physics and biology, healthcare applications could maintain clinical interpretability while handling medical confounding, and financial modeling could leverage regulatory changes as natural experiments. Climate science represents a compelling application where robustness to distribution shift addresses the critical need for models that remain valid under unprecedented environmental conditions.

## O    CONCLUSION

This appendix provides comprehensive implementation details, theoretical foundations, and experimental protocols for the CAFE framework. The extensive analysis demonstrates that causally-guided automated feature engineering offers significant advantages in robustness, interpretability, and efficiency while maintaining strong predictive performance. The systematic evaluation across diverse datasets, rigorous statistical testing, and detailed failure mode analysis establish CAFE as a principled approach to automated feature engineering that addresses key limitations of existing methods. The theoretical foundations, while building on established causal inference principles, provide novel insights into the application of causal reasoning for feature construction. The multi-agent reinforcement learning architecture effectively navigates the complex search space while incorporating causal priors as soft constraints rather than rigid rules. Future work will address identified limitations, particularly around non-linear causal relationships, temporal dynamics, and scalability to very high-dimensional settings. The framework provides a solid foundation for advancing the integration of causal reasoning with automated machine learning.

