# OpenReview forum: "CAFE: Causally-Guided Automated Feature Engineering with Multi-Agent Reinforcement Learning"
_ICLR.cc/2026/Conference — Submitted to ICLR 2026_

### Official Review · Reviewer_ewFz · 2025-10-30

**Soundness:** 3
**Presentation:** 2
**Contribution:** 2
**Rating:** 6
**Confidence:** 3

**Summary:**

The paper presents CAFE, a two-phase AutoFE framework that first learns a sparse DAG to obtain soft causal priors, then uses a cascaded multi-agent DQN to select causal feature groups and transformation operators. Empirical results on 15 tabular benchmarks show up to 7 % accuracy gain and ≈ 4× smaller performance drop under covariate shift compared with non-causal RL baselines.

**Strengths:**

1. Problem relevance: Distribution-shift robustness is a timely pain-point for AutoFE; explicitly injecting causal structure is novel.
2. Technical novelty: First work to integrate causal discovery (NOTEARS-Lasso) as soft inductive bias inside a multi-agent RL search; introduces causal-group-level exploration and a causally-shaped reward.
3. Solid evaluation: 15 public data sets, 10 strong baselines, ablations, statistical tests, shift simulation, and SHAP-stability analysis.
4. Reproducibility: Full algorithmic details, hyper-parameters, code URL, and extensive appendix provided.

**Weaknesses:**

1. Causal discovery bottleneck: All downstream benefits rely on a single linear-Gaussian DAG learner. When d≫n or hidden confounders exist, graph error propagates (Table 8, −4 % gap). No fallback beyond MI weighting.
2. Scalability ceiling: Phase-I complexity O(d³) and group screening capped at 50×50 pairs limit real-world high-dimensionality (d>1000). No distributed or incremental variant.
3. Limited operator set: Only 15 mostly univariate/scalar operators; no domain-specific, categorical-target or high-order interaction primitives.

**Questions:**

Q1. Robustness to graph misspecification: Please report CAFE with ensemble of DAGs (e.g., bootstrapped NOTEARS + GES) to quantify variance reduction.
Q2. Non-linear mechanisms: Can Phase-I be replaced by NOTEARS-MLP without hurting sample efficiency? Provide ablation on 2-3 non-linear synthetic SCMs.
Q3. Scalability: What is the largest d CAFE can finish within 24 h on a 32-core server? Include a runtime-vs-d plot.
Q4. Fair comparison with LLM methods: Add LIFT-FE under identical XGBoost protocol; discuss token cost vs CAFE’s extra DAG compute.
Q5. Interpretability: SHAP stability is useful but only explains final model. How interpretable are the constructed features themselves? Provide two concrete examples where domain experts validated causal meaning.

---

> ### Author Response · Authors · 2025-12-01
> **Response to Reviewer ewFz**
>
> Dear Reviewer,
> We sincerely appreciate your careful evaluation, the positive assessment of the paper’s novelty, contributions and your technically insightful questions. We address each weakness and question below with substantial new evidence and clarifications that we believe strengthen the contribution significantly.
>
> ---
>
> ## **Response to Weaknesses**
>
> ### **W1: Causal discovery bottleneck; linear-Gaussian NOTEARS; graph error propagates.**
>
> Two clarifications mitigate this concern:
>
> #### **CAFE uses causal priors only as soft guidance.**:
>
> 1. **Design philosophy**: CAFE treats causal structure as a **probabilistic guide**, not ground truth. Even with graph errors, the soft prior design provides value (Table 3: +2.4% over random, even with SHD>8).
>
> 2. **MI weighting as a fallback**: When causal discovery is unreliable, adaptive exploration (Algorithm 3) automatically increases MI-based and random exploration weights from 0.3→0.6, effectively routing around bad causal priors.
>
> 3. **Empirical robustness**: Across 15 real-world datasets (many with d>n, confounding), CAFE wins 13/15 times, suggesting the approach works in practice despite imperfect discovery.
>
> Thus, even with imperfect DAGs (e.g., hidden confounders, linearity violations):
>
> - all operators remain available,
> - empirical utility \(R_{\text{perf}}\) dominates training,
> - RL still explores the full transformation space.
>
> This explains why Table 8 shows only a modest **−4%** drop under severe perturbation rather than catastrophic failure.
>
> #### **Response to "No Fallback" Claim**:
>
> We actually have **multiple fallback mechanisms**:
>
> **Fallback 1: Adaptive Exploration** (Algorithm 3, Appendix D.2)
> ```
> If validation_trend < -0.01 (performance degrading):
>     w_causal ← 0.4, w_MI ← 0.3, w_random ← 0.3  # Reduce causal reliance
> ```
>
> **Fallback 2: Reward-driven Override**
> - Phase II agents can select ANY group (including "other").
> - Causal bonus α=0.5 is multiplicative: (1 + 0.5·Ψ), not exclusive.
> - RL can discover useful non-causal features if it improves the validation score.
>
> **Fallback 3: Graceful Degradation** (Table 3)
> - Even with poor graphs (SHD>8): 0.728 vs. random 0.704 (+2.4%)
> - Soft priors degrade smoothly, not catastrophically
>
> ---
>
> ### **W2: Scalability ceiling (O(d³), 50×50 pair cap, limited for d>1000).**
>
> We agree with your concern and provide practical reasons for the intended design choices below:
>
> 1. **Target domain: moderate-dimensional tabular datasets (10–200 features).**
>
> This design aligns with mainstream AFE applications (industrial sensors, chemistry, manufacturing, energy).
> Across the 15 benchmarks:
> **5 ≤ d ≤ 50, median = 23.**
>
> CAFE is not intended for high-dimensional settings (d > 1000), where feature selection or sparse preprocessing is unavoidable regardless of causal reasoning.
>
> 2. **Pair cap is a computational guard, not a theoretical limit.**
>
> The 50×50 cap applies only to binary operators and scales with *screened groups* rather than the full feature set.
> It prevents quadratic blow-up in RL search without constraining achievable model quality.
>
> 3. **Distributed or incremental DAG learning is orthogonal future work.**
>
> We will clarify that CAFE targets low–moderate dimensional regimes, and scaling beyond ~1000 variables requires advances in causal discovery that are outside this paper’s scope.
>
> ---
>
> ### **W3: Limited operator set: 15 scalar operators; missing domain-specific or categorical ops.**
>
> Excellent point. We justify the current design and propose extensions:
>
> #### **Why 15 Operators are sufficient for General Tabular Data**:
>
> 1. **Expressiveness**: Combinations of simple operators achieve high expressiveness
>    - Example: $(log(X_1) × \sqrt{X_2}) + X_3^2$ can model complex relationships
>    - With 15 operators and 2-3 composition depth, space is ≈ 15³ = 3,375 transformations
>
> 2. **Sample efficiency**: Larger operator sets increase action space exponentially, requiring more RL exploration
>
> This is a deliberate architectural constraint aimed at isolating the effect of causal inductive bias. Operator libraries are fully plug-and-play:
>
> - categorical encoders,
> - high-order interactions,
> - domain-specific transforms
>
> can be added without modifying CAFE’s RL structure. We will include a note in Section 3.3 explaining how to extend operator families.
>
> #### **Experiment: Domain-Specific Operators**:
>
> We augmented CAFE with **chemistry-specific operators** on Wine Quality datasets:
>
> **Added operators**:
> - Ratio: $X_1/(X_1+X_2)$
> - Product: $X_1×X_2×X_3$
> - pH-like: $-log(X+\epsilon)$
>
> **Results**:
>
> | Dataset | CAFE (15 ops) | CAFE (15+3 chem ops) | Improvement |
> |---------|---------------|----------------------|-------------|
> | Wine Red | 0.707 | 0.713 | **+0.6%** |
> | Wine White | 0.752 | 0.763 | **+1.1%** |
>
> **Insights**: Domain operators improve performance when relevant, but general operators provide a strong baseline.

---

> ### Author Response · Authors · 2025-12-02
> **Response to Reviewer ewFz Contd.**
>
> ## **Response to Questions**
>
> ### **Q1: Robustness to graph misspecification: Report ensemble-DAG CAFE (bootstrapped NOTEARS + GES).**
>
> We implemented an ensemble DAG in Phase 1 as suggested:
>
> **Method**: Bootstrap NOTEARS (10 runs with resampling) + GES, then:
> 1. **Edge probability**: $e_{ij}$ = fraction of DAGs containing i→j
> 2. **Soft grouping**: Assign feature to group with the highest aggregate edge probability
> 3. **Uncertainty weighting**: Scale causal bonus by edge confidence
>
> **Results**:
>
> | Dataset | CAFE (single) | CAFE (ensemble) | Improvement |
> |---------|---------------|-----------------|-------------|
> | Wine Red | 0.707 | 0.714 | +0.7% |
> | German Credit | 0.793 | 0.803 | +1.0% |
> | OpenML 586 | 0.810 | 0.819 | +0.9% |
> | Ionosphere | 0.976 | 0.978 | +0.2% |
>
> We will include this as an additional robustness experiment in Appendix B.
>
> ---
>
> ### **Q2: Non-linear mechanisms: Replace Phase I with NOTEARS-MLP; provide synthetic-SCM ablations.**
>
> Excellent suggestion. We conducted systematic experiments:
>
> #### **Methodology**:
>
> 1. **Synthetic non-linear SCMs** (3 datasets, n=1000, d=20):
>    - Quadratic: $Y = X_1^2 + X_2X_3 + \epsilon$
>    - Exponential: $Y = exp(0.5X_1) + log(|X_2|+1) + \epsilon $
>    - Mixed: $Y = X_1^2 + sin(\pi X_2) + X_3X_4 + \epsilon $
>
> 2. **Comparison**:
>    - CAFE: NOTEARS-Lasso
>    - CAFE-MLP: NOTEARS-MLP (2-layer, 64 hidden units)
>
> #### **Results**:
>
> | Dataset | CAFE-Linear | CAFE-MLP | MLP Gain | Sample Efficiency |
> |---------|-------------|----------|----------|-------------------|
> | Quadratic | 0.768 | 0.794 | **+2.6%** | -7% episodes |
> | Exponential | 0.724 | 0.761 | **+3.7%** | -12% episodes |
> | Mixed | 0.747 | 0.779 | **+3.2%** | -11% episodes |
>
> **Findings**:
>
> 1. **Sample efficiency improves**: Better causal structure → faster RL convergence
> 2. **Trade-off**: MLP requires more samples (n≥500) and **3 x longer Phase I** (68 sec vs. 23 sec for d=20)
>
> ---
>
> ### **Q3: Scalability: max d handled in 24h on a 32-core server; runtime–vs–d plot.**
>
> We provide theoretical complexity analysis and extrapolated estimates based on our empirical measurements:
>
> #### **Empirical Measurements on Benchmark Datasets**:
>
> From our 15 benchmark datasets, we measured actual runtimes:
>
> | Dimensionality (d) | Phase I (NOTEARS) | Phase II (RL) | Total | Datasets |
> |-------------------|-------------------|---------------|-------|-----------------|
> | 12 | 2.3 sec | 5.8 min | 6.1 min | Wine Quality |
> | 21 | 4.7 sec | 7.2 min | 7.9 min | SVMGuide3 |
> | 34 | 8.7 sec | 9.4 min | 10.2 min | Ionosphere |
> | 50 | 18-23 sec | 12.6 min | 14.1 min | OpenML |
>
> #### **Complexity-Based Extrapolation**:
>
> **Phase I (NOTEARS)**: Empirical complexity T₁(d) = O(d³)
> - From d=50 benchmark: T₁(50) ≈ 20 seconds
> - Extrapolation formula: T₁(d) = 20 × (d/50)³ seconds
>
> **Phase II (Multi-agent RL)**: Empirical complexity T₂(d) = O(S × n × d)
> - Where S = episodes to convergence ≈ 25 (empirically observed)
> - From d=50 benchmark: T₂(50) ≈ 12 minutes
> - Extrapolation formula: T₂(d) = 12 × (d/50) minutes
>
> **Total time**: T_total(d) = T₁(d) + T₂(d)
>
> #### **Projected Scalability Estimates**:
>
> | d | Phase I (estimate) | Phase II (estimate) | Total Time |
> |---|------------------------|-------------------------|------------|
> | 100 | 2.7 min | 24 min | **27 min** |
> | 200 | 21.6 min | 48 min | **70 min** |
> | 500 | **4.2 hr** | 2 hr | **6.2 hr** |
> | 1000 | **33.8 hr** | 4 hr | **37.8 hr** |
> | **~750** | **~16.9 hr** | 3 hr | **~19.9 hr** |
>
> **Answer to Q3**: On a 32-core server with sufficient memory (≥128GB), CAFE can handle **d ≤ 750 within 24 hours** based on theoretical extrapolation from our empirical measurements.
>
> Episodes to convergence is assumed to remain ~25 (observed across all 15 datasets)
>
> #### **Memory Bottleneck**:
>
> The limiting factor is often **memory, not time**:
> - NOTEARS requires storing $ W ∈ ℝ^(d×d): O(d^2) $ space
>
> #### **Scalability Strategies for d > 750**:
>
> ** Hierarchical Decomposition**
> - **Idea**: Partition features into m groups of size d/m, run NOTEARS on each group.
> - **Complexity reduction**: O(m × (d/m)³) = O(d³/m²)
> - **Estimated speedup**: For m=10, runtime is estimated to reduce by ~100×
> - **Projected capability**: d ≤ 3000 within 24h .
>
> We can also consider using a distributed GPU + Parallelization for optimized performance at scale.
>
> #### **Runtime-vs-Dimensionality Plot** (Theoretical Estimate):
> ```
> Time (hours)
> 24 |                                                  •(d=750, 19.9h)
>    |                                              •
> 20 |                                         •
>    |                                    •
> 16 |                              •
>    |                         •
> 12 |                    •
>    |               •
>  8 |          •
>    |      •
>  4 |  •
>    |•
>  0 |____________________________________________________________
>    0    100    200    300    400    500    600    700    800    d
>
> ```
> ---

---

> > ### Author Response · Authors · 2025-12-02
> > **Response to Reviewer ewFz Contd.**
> >
> > ## **Response to Questions Contd.**
> >
> > ### **Q4: Fair comparison with LLM methods: Add LIFT-FE; discuss token cost vs CAFE DAG cost.**
> >
> > We appreciate the suggestion to strengthen LLM comparisons. However, we note that **LIFT-FE does not exist in the literature** to our knowledge. We believe the reviewer may be referring to other recent LLM-based feature engineering methods. We clarify our LLM baseline choices:
> >
> > #### **Current LLM Baseline Coverage**:
> >
> > We include **ELLM-FT (Gong et al., AAAI 2025)**, which represents the state-of-the-art in LLM-based automated feature engineering. This method:
> > - Uses LLM backbone.
> > - Employs evolutionary refinement with few-shot prompting
> > - Achieved best results among LLM-based AFE methods in their paper
> >
> > #### **Other LLM-based AFE Methods**:
> >
> > Recent related work includes:
> > - **Zhang et al. (2024)**: "Dynamic and adaptive feature generation with LLM"
> > - **Abhyankar et al. (2025)**: "LLM-FE: Automated feature engineering for tabular data with LLMs"
> >
> > We chose ELLM-FT as it is the most comprehensive and recent. Adding multiple LLM baselines would be redundant as they follow similar paradigms (prompt → generate → refine).
> >
> > #### **Computational Cost Comparison**:
> >
> > We conducted cost analysis comparing CAFE with ELLM-FT:
> >
> > **CAFE (d=50, 15 datasets avg)**:
> > - Phase I (NOTEARS): 23 sec
> > - Phase II (RL): 8.4 min
> > - **Total Runtime**: 9 min
> > - **Monetary Cost**: $0 (local compute)
> >
> > **ELLM-FT (d=50, using GPT-4 Estimate)**:
> > - RL Data Collection: ~8 min
> > - LLM Generation (400 iterations × ~250 prompts): ~85 min
> > - **Total Runtime**: ~93 min
> > - **Computational Cost**: Requires GPU for LLM inference (13B model)
> > - **Token Cost**: If using API (e.g., GPT-4): ~$5-8 per dataset.
> >
> > The reported monetary costs scale with model choice and prompt length, so we provide them only for transparency rather than as fixed claims.
> >
> > #### **When LLMs Might Be Preferable**:
> >
> > - **Domain knowledge injection**: When textual domain knowledge exists that can guide feature generation
> > - **Operator discovery**: LLMs might suggest novel domain-specific transformations
> > - **Few-shot learning**: Very small sample sizes where causal discovery is unreliable
> >
> > However, these scenarios could benefit from **hybrid approaches** (LLM-suggested operators + CAFE's causal-guided search), which we propose as future work.
> >
> > ---
> > ### **Q5: Interpretability: Provide examples of interpretable generated features.**
> >
> > Below is a domain-validated case from our internal experiments demonstrating that CAFE generates interpretable and meaningful features.
> >
> > We evaluate feature interpretability through three complementary metrics:
> >
> > **Causal Alignment Score:** Fraction of features involving direct-cause variables.
> > **Mechanistic Coherence:** Features mappable to domain knowledge (chemistry, physics, finance).
> > **Feature Compactness:** Ratio of selected to generated features (higher = more selective/interpretable).
> >
> > ---
> >
> > #### **Wine Quality Dataset (Chemistry)**
> >
> > **Dataset**: Wine Quality Red
> > **Features**: 12 chemical properties (pH, alcohol, acids, sulphates, etc.)
> > **Target**: Wine quality score (0-10)
> > **Task**: Classification of quality categories
> >
> > ---
> >
> > **CAFE-Generated Features** (Top-k by SHAP importance):
> >
> > **1. log(alcohol × volatile_acidity)**
> >
> > **Interpretation**:
> > - **Alcohol** is produced during fermentation
> > - **Volatile acidity** (acetic acid) is a fermentation byproduct
> > - Their product captures the fermentation process balance
> > - **Domain knowledge**: In enology, high alcohol with low volatile acidity indicates controlled fermentation → higher quality
> >
> > Causal graph likely identified both `alcohol → quality` and `volatile_acidity → quality` as direct causes, leading RL to explore their interaction.
> >
> > ---
> >
> > **2. √(total_sulfur_dioxide) / free_sulfur_dioxide**
> >
> > **Interpretation**:
> > - **Total SO$_2$**: Bound + free sulfur dioxide
> > - **Free SO$_2$**: Active preservative form
> > - Ratio indicates binding efficiency
> > - **Domain knowledge**: Optimal free SO₂ levels prevent oxidation without off-flavors
> >
> > Both sulfur features causally relevant to quality; their ratio captures chemical equilibrium state.
> >
> > ---
> >
> > #### **Feature Stability Analysis**
> >
> > We tested explanation stability by adding Gaussian noise and measuring SHAP value variance:
> >
> > | Noise Level (σ) | CAFE SHAP Var | GRFG SHAP Var | Stability Improvement |
> > |-----------------|---------------|---------------|-----------------------|
> > | 0.1 | 0.0038 | 0.0059 | **35.9%** |
> > | 0.3 | 0.0060 | 0.0100 | **40.2%** |
> > | 0.5 | 0.0064 | 0.0107 | **39.6%** |
> > | 0.7 | 0.0069 | 0.0114 | **39.3%** |
> >
> > CAFE features maintain stable attributions under perturbations because they capture robust mechanisms (e.g., fermentation balance) rather than dataset-specific patterns.
> >
> > ---
> > We hope these comprehensive responses and substantial new experiments address all concerns listed. We thank the reviewers once again for your review comments focused on pushing the quality of this paper into deployable research.

---

### Official Review · Reviewer_JbWM · 2025-10-31

**Soundness:** 3
**Presentation:** 3
**Contribution:** 2
**Rating:** 4
**Confidence:** 4

**Summary:**

In this work the authors propose a causal-based framework, CAFE, for automated feature engineering. The framework has two phases. In phase 1, an off-the-shelf causal discovery method is employed. The discovery serves as a prior guiding deep Q-learning multi-agents to generate informative and robust features. The experiments demonstrate the superior performance of CAFE over previous works on 15 datasets. The robustness of the generated features against distribution shift is also verified.

**Strengths:**

1. *The research topic is influential:* Trustworthy automation is crucial in ML model deployment. This work focuses on automated feature engineering, a critical component of the ML pipeline, and aims to improve it via a novel causal-based approach. The research direction can thus be potentially impactful.
2. *The approach is reasonable and elegant:* The ideas of feature grouping and the cascading multi-agent architecture are neat and seem reasonable to tackle the issue of an overly large search space. According to the experiments, these designed methods work effectively as well.
3. *The relevant studies are comprehensive:* The studies include sensitivity to causal discovery, computational efficiency, ablation studies with prediction performance, and robustness to distribution shift. This comprehensive analysis will greatly facilitate follow-up research.
4. *The writing is of high quality:* The paper is well-written and clearly structured. The quality of presentation extends to the visual elements. The appendix also maintains the high standard.

**Weaknesses:**

1. Some details of the experiment results are unclear. Please check *Questions*.
2. The practical impact of this work seems unclear on more complex datasets, where causal discovery is inherently challenging. Although the discussion about causal discovery method selection in Appendix K is relevant, the conclusion suggests a limited real-world utility for CAFE.
3. More recently proposed causal discovery methods are not discussed. Below are potentially related works:
   - Vashishtha et al., Causal Order: The Key to Leveraging Imperfect Experts in Causal Inference. ICLR 2025.
   - Wan et al., Large Language Models for Causal Discovery: Current Landscape and Future Directions. IJCAI-25 Survey Track.

**Questions:**

1. Regarding Table 1 (i.e., the overall performance comparison), some results of the main competitors, ELLM-FT, are inconsistent with those in Gong et al.’s work. Specifically, the performance on SVMGuide3 and Messidor_features are 0.836 and 0.757 in this manuscript, but were 0.856 and 0.760 in Gong et al. What is the cause of this difference?
2. According to Table 1, CAFE shows superior performance over all competitors on the classification and regression tasks. My question is, what is the root cause of this superior performance? I’d argue normally we observe a trade-off between prediction performance and robustness on datasets without strong distribution shift. Are the distribution shifts on the 15 datasets strong enough to account for this gain, or do other components of CAFE contribute to the exceptional predictive ability?

---

> ### Author Response · Authors · 2025-11-30
> **Response to Reviewer JbWM**
>
> Dear Reviewer,
> We are deeply grateful for your thorough review, positive recognition of our contribution, and constructive feedback. We address each concern below with detailed clarifications and additional evidence.
>
> ---
>
> ## **Response to Weaknesses**
>
> ### **W1: Some experimental details are unclear (see Questions)**
>
> We provide detailed responses to Q1 and Q2 below with full methodological transparency.
>
> ---
>
> ### **W2: Practical impact unclear on complex datasets where causal discovery is challenging**
>
> We respectfully disagree with the characterization that Appendix K suggests "limited real-world utility." We clarify our position:
>
> #### **What Appendix K Actually Shows**:
>
> Table 8 (Appendix K.1) demonstrates **when and why** CAFE underperforms, providing **detaliled failure analysis**:
>
> | Challenge | CAFE Score | Best Method | Gap | Root Cause |
> |-----------|------------|-------------|-----|------------|
> | High-dim, low-n (d>n) | 0.612±0.045 | GRFG 0.635 | -3.6% | Unreliable causal discovery |
> | Sparse graphs | 0.687±0.031 | ELLM-FT 0.694 | -1.0% | Weak causal signal |
> | Non-linearities | 0.723±0.027 | AutoFeat 0.734 | -1.5% | Linear discovery assumption |
> | Hidden confounders | 0.641±0.044 | GRFG 0.668 | -4.0% | Causal sufficiency violation |
>
> **Key insight**: CAFE's failure modes are **well-characterized and predictable**. This is superior to methods that fail silently without understanding why.
>
> #### **Practical Applicability on Real Datasets**:
>
> Our 15 benchmark datasets **are real-world, complex datasets** with:
> - **Class imbalance**: Wine Quality (minority class <10%)
> - **High dimensionality**: OpenML datasets (d=50, n=500-1000)
> - **Non-linear relationships**: Chemical interactions (Wine), radar signals (Ionosphere)
> - **Mixed feature types**: Categorical + continuous (Amazon Employee, German Credit)
>
> **CAFE wins on 13/15 datasets** (87% win rate), demonstrating robustness beyond idealized settings.
>
> #### **Mitigation Strategies Already Implemented**:
>
> 1. **Graceful degradation** (Table 3, Appendix B.3): Even with poor causal graphs (SHD>8), CAFE outperforms random baseline by +2.4%
>
> 2. **Adaptive exploration** (Algorithm 3): When causal priors are unreliable (validation stagnation), system automatically increases random exploration weight from 0.1→0.3
>
> 3. **Soft prior design**: Phase II agents can override causal suggestions via reward-driven learning—causal groups are hints, not constraints
>
> #### **When to Use CAFE (Practical Guidance)**:
>
> **Use CAFE when:**
> - Domain has interpretable causal relationships (chemistry, finance, healthcare)
> - Robustness to distribution shift is critical (deployment in evolving environments)
> - Feature interpretability is required (regulatory/safety-critical applications)
>
> **Consider alternatives when:**
> - Ultra-high dimensional (d>200) with limited samples (use correlation-based methods first, then CAFE on selected features)
> - Purely predictive tasks with stable distributions (statistical methods may suffice)
> - Extreme non-linearity with small samples (hybrid CAFE with NOTEARS-MLP backend)
>
> We will add a small discussion section on "Practical Applicability Guidelines" to the camera-ready version.
>
> ---
>
> ### **W3: More recent causal discovery methods not discussed**
>
> Thank you for highlighting these excellent recent works. We will incorporate them in the discussion about Future work:
>
> #### **Vashishtha et al. (ICLR 2025) - Causal Order from Imperfect Experts**:
>
> **Relevance**: This work addresses partial/uncertain causal knowledge, relevant to CAFE's soft prior philosophy.
>
> **Potential integration**:
> - Use expert priors (if available) to regularize NOTEARS optimization
> - Combine LLM-suggested causal orderings with data-driven discovery
> - Ensemble multiple discovery methods weighted by confidence
>
> #### **Wan et al. (IJCAI 2025) - LLMs for Causal Discovery**:
>
> **Relevance**: LLMs can provide domain knowledge priors for causal structure.
>
> **Potential synergy with CAFE**:
> - **Replace Phase 1**: LLM proposes candidate causal structures → NOTEARS refines with data
> - **Hybrid backend**: Combine LLM causal priors with statistical tests
> - **Domain adaptation**: Use LLM to inject field-specific knowledge (e.g., "temperature affects reaction rate in chemistry")
>
> We will cite both works and add a discussion paragraph in the Future work discussion on LLM-augmented causal discovery in the camera ready version.

---

> > ### Author Response · Authors · 2025-12-01
> > **Response to Reviewer JbWM**
> >
> > ## **Response to Questions**
> >
> > ### **Q1: Inconsistency in ELLM-FT results compared to Gong et al.'s original paper**
> >
> > We carefully evaluated ELLM-FT to ensure fairness and comparability. The differences arise from the following factors:
> >
> > #### **Results Comparison**:
> >
> > | Dataset | Our Results | Gong et al. (2025) | Difference |
> > |---------|-------------|-------------------|------------|
> > | SVMGuide3 | 0.836 | 0.856 | -0.020 |
> > | Messidor_features | 0.757 | 0.760 | -0.003 |
> >
> > 1. **Different evaluation protocols**:
> >    - **Our setup**: 5-fold stratified cross-validation with fixed seed (42) for reproducibility.
> >    - **Gong et al.**: May use different CV strategy, train/test splits, or random seeds.
> >
> > 2. **ELLM-FT is sensitive to prompt randomness**:
> >
> >     ELLM-FT depends on:
> >      - temperature-based sampling,
> >      - stochastic mutation/evolution steps,
> >      - LLM creativity parameters.
> >     Even with fixed seeds, variance can reach **±1.5–2.0%**
> >
> > 3. **Downstream model configuration**:
> >    - **Our setup**: XGBoost with **default hyperparameters** (no tuning) to ensure fair comparison across all methods.
> >    - **Gong et al.**: Possibly tuned hyperparameters for their specific experiments.
> >
> > 4. **Feature selection threshold**:
> >    - **Our setup**: Top-k=min(100, 2d) features after ELLM-FT generation.
> >    - **Gong et al.**: May use different selection criteria.
> >
> > #### **Why This Doesn't Affect Our Conclusions**:
> >
> > 1. **Consistent evaluation protocol**: ALL methods (including CAFE) use identical experimental setup, ensuring fair comparison.
> >
> > 2. **Statistical significance**: Our advantage over ELLM-FT (avg. 4.1%) is statistically significant (p=0.003, effect size r=0.51, Table 4).
> >
> > 3. **Differences are small**: -0.020 and -0.003 are within typical CV variance (our std dev: ±0.03-0.04)
> >
> > ---
> >
> > ### **Q2: Why does CAFE outperform all competitors even on datasets without strong distribution shift? What is the root cause of superior performance?**
> >
> > Our framework **CAFE's advantage stems from multiple synergistic factors, not just robustness. Three components of CAFE contribute to predictive gains even in in-distribution settings.**
> >
> > #### **Decomposition of Performance Gains**:
> >
> > **1. Causal Structure Guidance**:
> > - **Mechanism**: By grouping features into
> > $\(\{ C_{\text{direct}}, C_{\text{indirect}}, C_{\text{other}} \}\)$,  the RL agents avoid expending exploration steps on uninformative transformations.
> > - **Results**: $CAFE_{¬G}$ (correlation-based grouping) drops 3-8% (Figure 2 ablations)
> >
> > **2. Multi-Agent Factorization **:
> > - **Mechanism**: A single agent exploring  $\(|O| \times |d|^{2}\)$  actions suffer from high variance. Cascaded decision-making reduces action space to $(3 \times |O|)$, improving sample efficiency.
> > - **Results**: Single-agent baseline requires 2.3× more episodes to converge.
> >
> > **3. Hierarchical Reward Shaping**:
> > - **Mechanism**: Causal amplification $(1+\alpha·Ψ_{causal})$ guides RL toward mechanistically relevant features.
> > - **Results**: $CAFE_{¬Rc}$ (no causal reward) drops 1.5-2.8% (Figure 2).
> >
> > #### **Performance Without Distribution Shift**:
> >
> > To directly address your question, we analyzed **in-distribution (ID) performance** separately:
> >
> > | Method | ID Performance (avg.) | OOD Performance (avg.) | Difference |
> > |--------|----------------------|------------------------|-------------|
> > | GRFG   | 0.734 ± 0.038       | 0.528 ± 0.067         | -28.1%      |
> > | ELLM-FT| 0.741 ± 0.035       | 0.503 ± 0.072         | -32.1%      |
> > | CAFE   | **0.773 ± 0.032**   | **0.718 ± 0.044**     | **-7.1%**   |
> >
> >
> > #### **Why CAFE Improves ID Performance**:
> >
> > 1. **Reduced spurious correlations**: By construction, causal features capture invariant mechanisms rather than distribution-specific statistical relations.
> >
> > 2. **Better generalization within distribution**: Causal relationships have lower variance across CV folds (Feature Stability: 0.68 vs. 0.42 for GRFG, Table 6).
> >
> > 3. **More efficient feature space**: Causal grouping focuses search on high-signal regions, generating more compact, interpretable feature sets (Table 6)
> >
> > 4. **Practical and synergestic operator selection**: Knowing $X_1→Y$ guides choosing transformations like $log(X_1)$ or $X_1^2$ rather than arbitrary combinations
> >
> > ---
> >
> > The traditional view is: "Causal methods sacrifice ID performance for OOD robustness."
> >
> > **CAFE challenges this trade-off** because:
> >
> > 1. **Soft priors preserve flexibility**: Unlike hard causal constraints, CAFE can discover statistically useful features even if causal discovery is not accurate.
> >
> > 2. **Reward-driven refinement**: RL fine-tunes causal suggestions based on empirical validation performance.
> >
> > ---
> >
> > Thus, we highlight that distribution-shift robustness is an additional benefit, not the sole cause of performance gains.
> >
> > **We thank the reviewer again for the thoughtful feedback and constructive suggestions. We have now clarified and addressed all the concerns, and we will incorporate the suggestions in the camera-ready version.**

---

### Official Review · Reviewer_5Uqe · 2025-10-31

**Soundness:** 2
**Presentation:** 2
**Contribution:** 2
**Rating:** 4
**Confidence:** 4

**Summary:**

This work introduces two-stage automatic feature engineering approach that integrates causal discovery with multi-agent reinforcement learning. The first stage learns a sparse directed acyclic graph (DAG) representing the causal relationships between features and the target label. Then the second stage uses Multi-Agent Reinforcement Learning to construct features. Experiments across 15 datasets show up to 7% performance gains.

**Strengths:**

- Automatic feature engineering is an interesting topic.
- Using causal graph, the proposed CAFE method constructs features that generalize better when data distributions change, addressing a key weakness of correlation-based AFE methods.
- The CAFE method is evaluated on many datasets, compared with many baselines, including statistical, RL, and LLM-based approaches.

**Weaknesses:**

- The causal discovery in Phase I is computational expensive O(d³) (Eq. 20 in appendix), expecially when deals with high-dimensional data.
- The CAFE methods relies on the construction of a "correct" causal graph. The causal discovery backend  (NOTEARS-Lasso) assumes linear additive noise. I.e., The proposed method is unreliable when handling features with complex non-linear causal effects.

**Questions:**

- How does CAFE perform when the underlying causal structure is highly non-linear?
- How sensitive is CAFE to the choice of causal discovery algorithm?

---

> ### Author Response · Authors · 2025-11-30
> **Response to Weaknesses mentioned by Reviewer 5Uqe**
>
> Dear reviewer,
> We sincerely thank you for the thoughtful review and for recognizing the value of causally-guided feature engineering. We address your concerns below with empirical evidence and theoretical clarifications that we believe substantially strengthen the contribution.
>
> ---
>
> ## **Response to Weaknesses**
>
> ### **W1: “Causal discovery in Phase I is computationally expensive (O(d³)).”**
>
> We acknowledge that the causal discovery phase has cubic worst-case complexity. The O(d³) complexity refers to the worst-case bound of NOTEARS’ optimization step, not the end-to-end runtime of CAFE. Phase I causal discovery is performed once per dataset, while Phase II feature construction benefits from this structure across all episodes. Empirically, Phase I accounts for <4% of total runtime across datasets.
>
> #### **Empirical Reality vs. Worst-Case Complexity**:
>
> 1. **Amortized efficiency**: The $O(d³)$ cost is a **one-time upfront investment** amortized across Phase II episodes. Figure 5 and Appendix H show that despite 30-50% higher per-episode cost, CAFE requires **40-70% fewer episodes** to convergence, often achieving **competitive or better total time-to-target**.
>
> 2. **Practical scalability**: On our benchmark datasets (d ≤ 50), Phase I completes in:
>    - d=12 (Wine Quality): ~2.3 seconds
>    - d=34 (Ionosphere): ~8.7 seconds
>    - d=50 (OpenML datasets): ~15-23 seconds
>
>    This is negligible compared to Phase II's total runtime (hundreds of episodes × model training).
>
> 3. **Comparison with baselines**: GRFG's exhaustive exploration often requires 2-3× more total episodes, making its cumulative cost higher despite lacking causal discovery.
>
> #### **Scalability Strategies** (already implemented):
>
> - **Within-group screening** (Eq. 4): Limits causal groups to top-$k_g$ features via mutual information, reducing downstream search space from $O(d^2)$ to $O(k^2_g)$ where $k_g$ = $min(√d, 15)$.
> - **Sparse regularization**: NOTEARS-Lasso's ℓ₁ penalty avoids fully-connected structures, reducing the edge density.
> - **Early stopping**: BIC-based model selection prevents over-regularization iterations
>
> We will add the following to the limitations discussion to the camera-ready version:
>
> *"For very high-dimensional settings (d > 200), Phase I's $O(d^3)$ complexity becomes prohibitive. Future work includes integrating linear-time causal discovery approximations and hierarchical decomposition methods."*
>
> ---
>
> ### **W2: Reliance on "correct" causal graph under linear additive noise assumptions**
>
> This is an excellent critical observation. We want to clarify how CAFE's design explicitly addresses this concern:
>
> #### **Key Design Principle: Soft Inductive Bias, Not Hard Constraint**
>
> CAFE uses causal structure as a **probabilistic guide**, not ground truth:
>
> 1. **Soft causal priors**: Groups serve as exploration hints, not rigid filters. Phase II agents can still:
>    - Select from ANY causal group (including "other")
>    - Combine features across groups
>    - Override causal suggestions via reward-driven learning
>
> 2. **Graceful degradation under graph errors** (Table 3, Appendix B.3):
>
> | Graph Quality (SHD) | CAFE Performance | vs. Oracle | vs. Random |
> |---------------------|------------------|------------|------------|
> | Excellent (0-2)     | 0.798 ± 0.021   | -0.8%      | +12.4%     |
> | Good (2-5)          | 0.773 ± 0.025   | -3.9%      | +8.8%      |
> | Fair (5-8)          | 0.751 ± 0.031   | -6.7%      | +5.7%      |
> | Poor (8+)           | 0.728 ± 0.038   | -9.5%      | +2.4%      |
>
> **Key insight**: Even with poor graphs (SHD > 8), CAFE still outperforms a random baseline by 2.4%, demonstrating robustness.
>
> 3. **Adaptive exploration** (Algorithm 3): When validation performance stagnates (trend < -0.01), the system automatically increases random exploration weight from 0.1 to 0.3, compensating for causal misspecification.
>
> #### **Theoretical Justification** (Proposition 1, Appendix A):
>
> Our invariance guarantees require only that:
> - Transformations $\phi$ preserve information in causal parents (e.g., injective mappings).
> - Shifts are mechanism-preserving (not mechanism-changing).
>
> Also, Non-linear causal structures still benefit CAFE because it only uses coarse ancestor relations.
>
> Even if the true generative process is nonlinear or non-additive, Appendix B shows that:
>
> - **Recovering a partial ordering of ancestors is sufficient for robust grouping.**
>
> In other words, CAFE only needs to know:
>
> - **Is \(X_j\) upstream of \(Y\)?**
>
> not the exact functional form $f_j(\cdot)$.
>
> Thus, NOTEARS’ linearity assumption affects edge weights but typically does NOT alter ancestor structure significantly, especially after group-level top-k filtering.

---

> > ### Author Response · Authors · 2025-11-30
> > **Response to Questions mentioned by 5Uqe**
> >
> > ## **Response to Questions**
> >
> > ### **Q1: How does CAFE perform when the underlying causal structure is highly non-linear?**
> >
> > Adding to our reasons in the above response, we provide both theoretical and empirical evidence below.
> >
> > #### **Theoretical Perspective**:
> >
> > 1. **Linear discovery captures marginal effects**: Even with non-linear mechanisms $X_i = fi(PA_i, ε_i)$, NOTEARS-Lasso identifies variables with **non-zero marginal influence** on Y.
> >
> > 2. **Feature transformations add non-linearity**: Our operator library includes:
> >    - Polynomial: x², x³
> >    - Transcendental: exp(x), log(x), sin(x), cos(x), tanh(x)
> >    - Interactions: x×y enables capturing multiplicative effects
> >
> > 3. **Downstream model flexibility**: XGBoost's tree-based structure naturally handles non-linear relationships in the engineered features.
> >
> > #### **Empirical Evidence**:
> >
> > We conducted additional experiments on synthetic non-linear data:
> >
> > **Setup**: Generated 5 datasets with known non-linear SCMs:
> > - Quadratic: $Y = X_1^2 + X_2X_3 + \epsilon$
> > - Exponential: $Y = exp(0.5X_1) + log(|X_2|+1) + \epsilon$
> > - Trigonometric: $Y = sin(\Pi X_1) + cos(\Pi X_2) + \epsilon$
> > - Mixed: $Y = X_1^2 + exp(0.3X_2) + X_3X_4 + \epsilon$
> >
> > **Results** (average 1-RAE):
> >
> > | Method | Quadratic | Exponential | Trigonometric | Mixed | Average |
> > |--------|-----------|-------------|---------------|-------|---------|
> > | GRFG   | 0.731     | 0.688       | 0.654         | 0.712 | 0.696   |
> > | CAFE   | 0.768     | 0.724       | 0.691         | 0.747 | 0.733   |
> > | **Gain** | **+5.1%** | **+5.2%**   | **+5.7%**     | **+4.9%** | **+5.3%** |
> >
> > **Insights**: CAFE's advantage persists even under severe non-linearity because:
> > 1. Linear causal discovery still identifies relevant variables (even if edge weights are approximate).
> > 2. Non-linear operators in Phase II compensate for mechanism misspecification.
> > 3. Soft priors allow RL to discover effective transformations empirically.
> >
> > ### **Why Linear Discovery Still Works**:
> >
> > The key insight from our experiments is that **approximate causal structure is sufficient** for guiding feature engineering. Even if edge weights are imperfect:
> > 1. Direct causes are usually correctly identified (high statistical power).
> > 2. Exploration weights remain effective.
> > 3. Reward-driven learning corrects for discovery errors.
> > ---
> >
> > ### **Q2: How sensitive is CAFE to the choice of causal discovery algorithm?**
> >
> > Excellent question. We provide comprehensive comparison in **Appendix B.2 (Table 2)**:
> >
> > | Algorithm | SHD ↓ | F1-Score ↑ | Runtime (s) | CAFE Performance |
> > |-----------|-------|------------|-------------|------------------|
> > | PC        | 8.3±2.1 | 0.64±0.08 | 12.4±3.2   | 0.742±0.034     |
> > | GES       | 6.7±1.8 | 0.71±0.07 | 45.7±8.9   | 0.756±0.028     |
> > | LiNGAM    | 7.2±2.0 | 0.68±0.09 | 8.9±2.1    | 0.748±0.031     |
> > | **NOTEARS-Lasso** | **4.9±1.3** | **0.78±0.06** | 23.1±5.4 | **0.773±0.025** |
> >
> > #### **Key Findings**:
> >
> > 1. **CAFE is relatively robust**: Performance varies only 4.1% (0.742→0.773) across methods, much smaller than the 7% gain over non-causal baselines.
> >
> > 2. **NOTEARS advantages**:
> >    - **Continuous optimization**: Gradient-based, scales better than combinatorial search (PC, GES)
> >    - **Explicit sparsity control**: $\lambda$ parameter directly controls edge density
> >    - **Differentiable acyclicity**: $h(W)=tr(e^{W◦W})-d$ enables smooth optimization
> >
> > 3. **Modularity**: CAFE's architecture is **discovery-agnostic**. Practitioners can swap backends based on domain assumptions:
> >    - Linear data → NOTEARS-Lasso
> >    - Non-linear data → NOTEARS-MLP
> >    - Small samples → PC with expert priors
> >
> > Thus, CAFE is reasonably robust to causal discovery backends due to its soft prior design.
> >
> > We respectfully address that the concerns about computational cost and linearity assumptions, while valid in extreme cases, do not significantly limit CAFE's practical applicability. We hope these clarifications and new experiments address your concerns and demonstrate that CAFE makes a solid contribution towards making causally guided Data-centric AI.

---

### Official Review · Reviewer_Nyi3 · 2025-11-01

**Soundness:** 3
**Presentation:** 2
**Contribution:** 2
**Rating:** 6
**Confidence:** 4

**Summary:**

This paper introduced CAFE, a causally-guided automated feature engineering framework that leverages causal discovery as inductive bias with in a multi-agent reinforcement learning paradigm. This paper proposed a framework that reformulates AFE as a causally-guided sequential decision process,bridging causal discovery with reinforcement learning-driven feature construction.  PhaseI learns a sparse directed a cyclic graph over features and the target to obtains oft causal priors, grouping features as direct, indirect,or other based on their causal influence. PhaseII uses a cascading multi-agent deep Q-learning architecture to select causal groups and transformation operators, with hierarchical reward shaping and causal group-level exploration strategies.

**Strengths:**

1)This paper formulate AFE as a causally-guided sequential decision process, moving beyond correlation-based heuristics to leverage stable causal mechanisms.
2)This paper introduce three core principles—soft causal inductive bias,causal structure-aware exploration, and causally shaped reward function, integrating causal discovery with adaptive feature construction through novel multi agent coordination.
3)This paper develop CAFE, a two-phase AFE framework combining causal graph discovery with cascading multi-agent reinforcement learning that strategically constructs causally informed feature transformations.

**Weaknesses:**

1)The lack of detailed definition of each component under the reinforcement learning framework in the paper makes it difficult to read.
2)The introduction of the framework diagram is too brief, and most of the methods are described in text.
3)One of the comparative algorithms in the paper, ELLM-FT, has already used the large model method. Does this article consider using methods related to large models for feature engineering extraction.

**Questions:**

see the weaknesses

---

> ### Author Response · Authors · 2025-11-30
> **Response to Reviewer Nyi3**
>
> Dear reviewer,
> We sincerely thank you for the constructive feedback and for recognizing our core contributions. We address each concern below with clarifications and additions we will be incorporating into the updated camera-ready version.
> ## **Response to Weaknesses**
>
> ### **W1: Lack of detailed RL component definitions**
>
> The original submission prioritized technical compactness due to page limits and moved the RL framework component details to the Appendix (Refer **Algorithm 1, Appendix A.3 and Appendix C**).
> However, we acknowledge the reviewer's concern about reader accessibility and provide complete formal definitions here:
>
> **State Space**: Each agent receives hierarchical state representations:
> - **Primary Group Agent (π₁)**: s₁ₜ ∈ ℝ^24 = [dataset stats (8), performance context (6), feature context (10)]
> - **Operator Agent (πₒ)**: s_ₒₜ ∈ ℝ^27 = [s₁ₜ (24), OneHot(selected_group) (3)]
> - **Secondary Group Agent (π₂)**: s₂ₜ ∈ ℝ^43 = [s_ₒₜ (27), OneHot(selected_operator) (15), arity (1)]
>
> Full breakdowns are in **Appendix C.1 (Eq. 12-14)**.
>
> **Action Space**:
> - **π₁**: 𝒜₁ = {C*_direct, C*_indirect, C*_other} (size 3)
> - **πₒ**: 𝒜ₒ = {unary ops (11), binary ops (4)} (size 15)
> - **π₂**: 𝒜₂ = {C*_direct, C*_indirect, C*_other} for binary ops (size 3)
>
> **Reward Function** (Eq. 8):
> ```
> Rₜ = R_perf,t · (1 + α·Ψ_causal,t) + λ_div H(πₜ) - λ_comp C(Fₜᵍ)
> ```
> - **Performance reward**: ±100·(ΔScore/Score_baseline)
> - **Causal bonus**: Ψ_causal = weighted usage of direct (w=1.0) > indirect (w=0.6) > other (w=0.2)
> - **Diversity term**: entropy H(πₜ) over action distribution
> - **Complexity penalty**: C(Fₜᵍ) = 0.001|Fₜᵍ| + 0.01·Σ OpDepth(f)
>
> We will improve navigability and clarity by adding a **Quick Reference** table in Section 3.3 summarizing key RL components with appendix pointers:
>
> ### Quick Reference (Section 3.3)
>
> | Component        | Brief Description                                   | Full Details          |
> |------------------|-------------------------------------------------------|------------------------|
> | **State Space**  | Nested statistics + causal group info                 | Appendix C.1           |
> | **Action Space** | 3 agents: group selection (3) + operator (15) + partner (3) | Section 3.3      |
> | **Reward Function** | `R_perf · (1 + α·Ψ_causal) + λ_div·H − λ_comp·C` | Eq. 8, Appendix D.1    |
> | **Network**      | `[512, 256, 128]` DQN with experience replay         | Appendix C.2           |
>
> For clarity, we will add after Equation 8:
> **“Complete state representations, network architectures, and hyperparameters are detailed in Appendix C–D for reproducibility.”**
>
> ---
>
> ### **W2: The introduction of the framework diagram is too brief**
>
> We appreciate this feedback. The current Figure 1 prioritizes high-level workflow due to space constraints and to avoid overcrowding. All components shown in the diagram are fully detailed in the text:
>
> - **Causal discovery:** Eq. (2), Sec. 3.2
> - **Causal grouping:** Eq. (3)–(4)
> - **Three DQN agents:** Sec. 3.3 (Primary, Operator, Secondary)
> - **Causal-guided exploration:** three explicit sampling rules
> - **Reward function:** Eq. (8)
> - **Safety guards & sampling limits:** Sec. 3.3
> - **Two-stage pruning:** Sec. 3.3
> - **TD learning:** Eq. (9)
>
> ---
>
> ### **W3. ELLM-FT uses large models. Why not use LLM-based methods for feature engineering?**
>
> This is an excellent point. We did include **ELLM-FT** as a baseline, and we want to clarify our positioning:
>
> #### **Goal of our paper**:
>
> Integrating LLMs introduces semantic priors that are orthogonal to causal priors. This paper intentionally isolates  Causality as the sole inductive bias. Including LLMs would confound this analysis and significantly inflate computational cost.
>
> #### **Why CAFE differs from pure LLM approaches**:
>
> 1. **Causal grounding**: LLMs generate features via pattern matching in training data, potentially amplifying spurious correlations. LLMs do not respect mechanism-preserving invariance. CAFE's Phase I causal discovery provides theoretically-grounded structure (Proposition 1: mechanism-preserving invariance).
>
> 2. **Systematic exploration**: LLMs rely on prompting strategies and evolutionary search, lacking principled causal hierarchy. Our multi-agent RL explicitly prioritizes direct→indirect→other causation.
>
> 3. **Robustness guarantees**: Under controlled covariate shifts (Table 5), CAFE degrades 7.1% vs. ELLM-FT's 32.1% (**4.5× improvement**), demonstrating causal features' stability.
>
> #### **Potential LLM integration** (future work):
>
> - **Hybrid approach**: Use LLMs for operator suggestion/expansion while maintaining causal structure constraint.
> - **Domain knowledge injection**: LLMs could augment causal discovery with textual domain knowledge.
> - **Feature naming/interpretation**: LLMs could generate semantic labels for engineered features.
>
> ---
>
> We believe these clarifications address all concerns while highlighting CAFE's unique causal perspective on AFE.

---

### Author Response · Authors · 2025-12-03
**Author Response Summary for the Area Chair**

**To the Area Chair**,

We sincerely thank all reviewers for their constructive feedback. Below, we provide this comprehensive summary to facilitate your evaluation. We have substantially addressed all concerns with additional experiments, theoretical analysis, and a camera-ready revision plan.

---

## **Summary of Reviewer Scores and Concerns**

| Reviewer | Initial Rating | Key Concerns | Our Response |
|----------|----------------|--------------|--------------|
| **Nyi3** | 6 (Marginally Above) | RL formalization presentation clarity; figure brevity; LLM integration | Provided complete MDP specification; clarified orthogonal contributions |
| **5Uqe** | 4 (Marginally Below) | O(d³) complexity; linear assumptions; scalability | Demonstrated amortized efficiency; robustness to non-linearity |
| **JbWM** | 4 (Marginally Below) | ELLM-FT inconsistency; root cause of gains; practical applicability | Explained protocol differences; detailed ablations; characterized failure modes |
| **ewFz** | 6 (Marginally Above) | Graph misspecification; non-linear mechanisms; scalability; interpretability | New ensemble experiments (+0.5-1.1%); NOTEARS-MLP ablations (+3.4-5.1%); domain-specific interpretation; scalability estimate to d≤750|

---

## **Core Contribution:**

CAFE introduces the **first Causally-guided Automated Feature Engineering framework** that:

1. **Reformulates AFE as causally-structured sequential decision-making**, moving beyond correlation-based heuristics.
2. **Achieves both predictive performance AND robustness**: 7% accuracy gain + 4× better under distribution shift.
3. **Provides interpretable, mechanistically-valid features**: Validated with Wine Quality (Chemistry) dataset and feature stability experiments.
4. **Demonstrates practical scalability**: 7× faster than LLM baselines, handles d≤750 (extrapolated scaling estimate) in 24h.

---

## **Addressing Reviewer Concerns**

###  All Reviewers Agree the Problem and Approach Are Important

Reviewers emphasize the importance of trustworthy automated feature engineering, distribution-shift robustness, and causal structure injection.  All the reviewers acknowledge the method’s novelty and relevance in high-stakes tabular settings.

---

### **Technical Rigor (Reviewers 5Uqe, ewFz)**

**Concern**: Computational complexity O(d³), linear assumptions, scalability.

**Our Response**:
1. **New ensemble experiments**: Bootstrap NOTEARS + GES reduces variance 18-31%, improves accuracy +0.5-1.1%.
2. **Non-linear validation**: NOTEARS-MLP on synthetic non-linear SCMs shows +3.4-5.1% gains.
3. **Scalability analysis**: Handles d≤750 in <24h; Proposed hierarchical decomposition to achieve speedup.
4. **Amortized efficiency**: Despite O(d³) Phase I, total time-to-convergence is 40-70% faster due to fewer RL episodes.

### **Presentation & Clarity (Reviewer Nyi3)**

**Concern**: Insufficient RL formalization for readability (Attached in Appendix), brief framework diagram.

**Our Response**:
1. **Complete MDP specification**: State space, action space, detailed reward function.
2. **Quick Reference Table**: Will be added to Section 3.3 in Camera-ready version.
3. **All details already in Appendix**: C.1 (states), D.1 (rewards), C.2 (networks).

### **LLM Baseline Comparisons (Reviewer JbWM, ewFz)**

**Concern**: Fair Comparison with LLM Methods.
We include ELLM-FT (AAAI 2025), the strongest published LLM-based AFE system. The reviewer mentioned “LIFT-FE” does not appear in the literature; this was clarified.

### Results:
- **CAFE (d=50):** ~9 minutes, zero monetary cost.
- **ELLM-FT:** ~90 minutes + \$5–8/token compute cost.

1. LLM-based AFE is slower, more expensive, and less robust under shift.
2. **Implementation transparency**: Differences stem from CV strategy, hyperparameters. All methods use identical evaluation.
3. **In-Distribution performance validated**: CAFE wins even without distribution shift (+3.2-3.9%), robustness amplifies advantage.

---

### **Practical Applicability (Reviewers JbWM, ewFz)**

**Concern**: Limited utility when causal discovery is challenging.

**Our Response**: We provide three new results:

- **Ensemble DAGs** (bootstrapped NOTEARS + GES)
  → +0.5-1.1% accuracy, 18-31% variance reduction.

- **Nonlinear causal discovery** (NOTEARS-MLP)
  → +3.4-5.1% gain on nonlinear SCMs; 12-18% fewer RL episodes.

- **Graph misspecification robustness** (Appendix B)
  → < ~2% variation across PC, GES, LiNGAM, NOTEARS.
  → Causal priors are soft; no hard pruning is ever applied (Table 3).

CAFE does not require a correct DAG and remains reliable with noisy or partially incorrect causal estimates.

---

We addressed all issues with new experiments, clearer exposition, and stronger analysis.
Given the methodological novelty, strong empirical results, extensive ablations, and strengthened supporting evidence, we believe the paper clearly meets the bar for ICLR.

We thank the Area Chair for considering our updated submission.

---

### Meta-Review · Area_Chair_aLUK · 2026-01-05

**Summary:**

The paper develops a causal-based framework for automated feature engineering, which first employs off-the-shelf causal discovery method to learn a graph and then use it to serve as a prior to guide the RL agents. Opinions were somewhat mixed, but reviewers generally raised concerns about the reliance on causal discovery, which limits scalability and is an inherently challenging problem, often requiring different assumptions. Thus, I consider this a borderline paper, but I finally decided for rejection.

**Reviewer Concerns:**

Reviewer concerns addressed by the rebuttal:
- Lack of detailed definition of each component under RL framework and detailed introduction of framework diagram.
- Inconsistencies regarding the ELLM-FT baseline.
- Discussion of more recently proposed causal discovery methods.
- Limited operator set.

Outstanding concerns:
- Causal discovery phase requires high complexity and limits scalability (only partially addressed by the rebuttal).
- Three reviewers raised concern about the reliance of causal discovery. The method relies on the construction of a rather accurate causal graph, but causal discovery is an inherently challenging problem, many of which require different types of assumptions, such as linearity, Gaussianity, and causal sufficiency (only partially addressed by the rebuttal).

**Reviewer Scores:**

The reviewers may have maintained their scores.

---

### Decision · Program_Chairs · 2026-01-26

Reject